

# Elucidating ice formation pathways in the aerosol-climate model ECHAM6-HAM2

Remo Dietlicher[1], David Neubauer[1], and Ulrike Lohmann[1]

[1]Institute for Atmospheric and Climate Science, ETH Zürich, Universitätstrasse 16, 8092 Zürich, Switzerland.

*Correspondence to:* Remo Dietlicher (remo.dietlicher@env.ethz.ch)

**Abstract.** Cloud microphysics schemes in global climate models have long suffered from a lack of reliable satellite observations of cloud ice. At the same time there is a broad consensus that the correct simulation of cloud phase is imperative for a reliable assessment of Earth's climate sensitivity. Combining new satellite products (from CloudSat and CALIPSO) and physically-based ice microphysics parameterizations allows for rapid progress in reducing the inter-model spread in predicting

the cloud phase partitioning at sub-zero temperatures. This work introduces a new method to build a sound cause-and-effect relation between the microphysical parameterizations employed in our model and the resulting cloud field through a quantitative cloud formation pathway analysis. We find that heterogeneous freezing in super-cooled liquid clouds only dominates ice formation in roughly $7\,\%$ of the simulated cloud volume, a small fraction compared to almost $65\,\%$ of the cloud volume governed by homogeneous freezing below $-35\,°\mathrm{C}$. Compared to the CALIPSO-GOCCP satellite product, our model overestimates the

relative frequency of occurrence of cloud ice in the mixed-phase temperature regime. The ice formation pathway analysis reveals that this is caused by too much cloud ice propagating from the cirrus into the mixed-phase cloud regime, an unexpected result. This suggests that further efforts to improve the cloud phase partitioning must target cloud overlap assumptions for sedimentation and the related below cloud sublimation.

## 1 Introduction

Clouds are an important modulator for Earth's climate. They excert a net radiative forcing of approximately $-20\,\mathrm{W\,m^{-2}}$ and thus significantly cool the planet (Boucher et al., 2013). Compared to this, the forcing induced by well-mixed greenhouse gases since pre-industrial times is almost one order of magnitude smaller, approximately $3\,\mathrm{W\,m^{-2}}$ (Myhre et al., 2013), and has the opposite sign associated with a warming. Small changes in the cloud radiative forcing can therefore easily offset or strengthen a greenhouse gas induced warming. In fact, there is a broad consensus that clouds contribute the largest uncertainty for climate

projections in state-of-the-art general circulation and Earth system models (Flato et al., 2013).

    The models that contributed to the Community Model Intercomparison Project Phase 5 (CMIP5) generally agree that cloud adjustments to a warming climate will likely further reinforce the initial warming, a positive feedback loop. Clouds are expected to become fewer, higher and optically thicker in the global mean (Zelinka et al., 2013).

    The warming response of cloud ice has been hypothesized to counteract and therefore reduce the global mean, net positive

cloud feedback through a transition from optically thin ice clouds in the present day climate to optically thick liquid clouds





in a warmer climate (Mitchell et al., 1989; Storelvmo et al., 2015; Ceppi et al., 2016; Frey and Kay, 2018). The magnitude of this so-called cloud phase feedback strongly depends on the simulation of the present-day cloud phase partitioning in models. It is well established that models tend to underestimate the ratio of supercooled liquid water to ice in present-day conditions (Cesana et al., 2015). This leads to an overestimation of the magnitude of the cloud phase feedback (Li and Le Treut, 1992; Terai et al., 2016; Tan et al., 2016) in a greenhouse induced, warmer climate in some models. Recent modeling efforts challenge the universality of this finding (Lohmann and Neubauer, 2018; Bodas-Salcedo, 2018), calling for a comprehensive description of ice formation pathways in GCMs.

Especially sensitive to global warming is ice in mixed-phase clouds, predominantly originating from heterogeneous immersion or contact freezing of mineral dust (Atkinson et al., 2013; DeMott et al., 2015; Kanji et al., 2017). Those clouds are usually found in the temperature range between $-38\,°C$ and $0\,°C$ which we therefore refer to as the mixed-phase temperature regime. Beyond the theoretical and experimental uncertainties of the freezing mechanisms (Welti et al., 2014; Ickes et al., 2015; Marcolli, 2017), a realistic representation of cloud glaciation is further complicated by uncertainties in the parameterization of subsequent ice growth. Due to the lower saturation water vapor pressure over an ice crystal compared to a liquid droplet surface, cloud ice can grow below water saturation, leading to evaporation of the cloud droplets. This process is called the Wegener-Bergeron-Findeisen (WBF) process and can glaciate a cloud within minutes to hours (Korolev and Isaac, 2003), depending on the number of ice crystals, temperature and vertical velocity of the air parcel.

Due to the complex processes governing mixed-phase clouds, models inhibit a very large spread in the simulation of the phase partitioning (Fan et al., 2011; McCoy et al., 2016). A model intercomparison study of Komurcu et al. (2014) found for the models in their study that even for the same ice nucleation parameterizations, the spread in the simulated phase partitioning among the models was not reduced.

A significant portion of cloud ice is found at temperatures below $-38\,°C$. There, ice crystals can freeze homogeneously from pre-activated cloud droplets (Lohmann et al., 2016) and deliquesced aerosols (Koop et al., 2000) or nucleate directly on an ice nucleating particle (INP). The latter two processes do not require water saturation, leading to fundamentally different types of cirrus clouds (Krämer et al., 2016; Wernli et al., 2016; Gasparini et al., 2018) with significant differences in the respective microphysical and thus optical properties, highlighting the need for a correct representation of ice formation pathways of all clouds containing ice, not just those in the mixed-phase regime.

In this study we employ an adapted version of the Predicted Bulk Particle Properties (P3) scheme of Morrison and Milbrandt (2015) for the use in a GCM. Single category ice phase microphysics schemes have recently also been implemented in the Community Atmosphere Model Version 5 (CAM5) GCM (Eidhammer et al., 2017; Xi et al., 2017). Dietlicher et al. (2018) (hereafter D18) provides a detailed description of the technical aspects of the implementation of P3 in ECHAM6-HAM2, the GCM we use in this study. This scheme has the decisive advantage of representing cloud ice in a consistent manner, predicting the particle size distribution as well as the mass-to-size relationship. It employs one single, prognostic ice phase category, rendering heuristic conversion rates between in-cloud and precipitation type ice hydrometeors unnecessary. Process rates are computed offline and read back from look-up tables. This allows to integrate each process rate over the entire particle spectrum. For example, our scheme consistently represents the size dependence of the depositional growth rate using a size-dependent



ice crystal capacitance. Simplified, size-independent ice crystal growth rates are hypothesized to significantly contribute to the large spread among models in the simulated phase partitioning (Fan et al., 2011).

As has been eluded to above, the formation history of a cloud plays a decisive role, both for mixed-phase and cirrus clouds. Traditional model output is heavily aggregated spatially and temporally. Therefore, information on the cloud formation path-

ways is lost and must be inferred retroactively from the aggregated model states. Cause-and-effect relations between the microphysical parameterizations and the resulting cloud state are blurred as a result, which is a problem for model development where such relations are essential to target faulty or unrealistic parameterizations. Unlike the real atmosphere which does not directly reveal its governing processes, the models do by definition. This retroactive inference is therefore unnecessary and in part defeats the purpose of model simulations: the forward integration of differential equations. Here we introduce a new

method to trace and quantify the microphysical cloud formation pathways and use cloud classification to compute cloud type statistics.

The benefit of cloud classification is well established and has a long history. A common method is the identification of dynamical cloud regimes to compute conditional statistics (Jakob and Tselioudis, 2003; Williams and Tselioudis, 2007; Williams and Webb, 2009; Tsushima et al., 2016). These methods use high-frequency output and short-term temporal aggregation of

cloud top temperature vs. cloud optical thickness histograms together with clustering algorithms, classifying the mean dynamic states over the aggregation period. Our method focuses on the microphysics of cloud formation linking parameterizations and the resulting cloud field. This link is essential to understand the simulated cloud field and can be used as an effective tool to identify deficient parameterizations.

This study is composed of two parts. In Sect. 2 we highlight the main differences in the model employed here compared

to the reference model version ECHAM6.3-HAM2.3 and discuss the new cloud fraction parameterization that has been added since the previous description of the new model in D18. We validate the simulation of clouds, with a special focus on cloud ice, against a series of satellite observations in Sect. 3. In the second part, we introduce additional prognostic equations that are solved to quantify cloud formation pathways in Sect. 4 which are used to compile a climatology of cloud types in Sect. 5 to define explicit targets for further progress in microphysical parameterizations.

## 2   Model description

The improvements to the previous model version ECHAM6.3-HAM2.3 (Neubauer et al., 2014)) (hereafter called the reference model) are twofold. In the new model, cloud ice is a fully prognostic quantity, i.e. the advection equation for vertical transport (sedimentation) is solved online. Describing all ice particles with only a single category based on Morrison and Milbrandt (2015) no longer requires the heuristic partitioning between in-cloud ice and precipitating snow. The technical details of the

model regarding prognostic treatment of sedimentation and the associated numerical stability restrictions on the time-step as well as an inter-comparison of the different ice representation schemes can be found in D18. Here we will focus on the global evaluation of the new model both in terms of feasibility of the results as well as computational performance.




The predicted bulk particle properties scheme (P3) presented in Morrison and Milbrandt (2015) predicts 4 properties of the ice particle distribution: the total ice mass mixing ratio $q_i$, the total ice number mixing ratio $N_i$, the rimed ice mass mixing ratio $q_{rim}$ and the rimed ice density $b_{rim}$. Originally developed for the Weather Research and Forecasting model (WRF), the scheme was not intended for the coarse resolution of GCMs and climate projections. As we have shown in D18 by idealized single column model (SCM) simulations, GCMs are likely unable to properly represent small-scale weather features like squall lines to produce significant rimed ice formation. We test this hypothesis in the global setup of ECHAM-HAM in this paper.

The single conceptual deviation from the microphysics scheme described in D18 concerns the sub-grid cloud cover parameterization. While this work focuses on the importance of the representation of cloud ice, the sub-grid cloudiness is a fundamental property of clouds in GCMs, governing both cloud-radiation and cloud-precipitation interactions.

The approach taken in D18 tried to minimize the difference between the new and the reference model by extending the approach of Sundqvist et al. (1989) (hereafter S89) to ice clouds with a smooth transition from liquid water to ice clouds. As we will discuss in Sect. 3, this scheme leads to a strong positive bias of cloud cover for cirrus clouds. Therefore, we used a slightly different cloud cover scheme here.

The cloud cover scheme has also implications for the growth by condensation and deposition. Diagnosing sub-grid cloud cover from the large-scale relative humidity creates a tight link between the cloud fraction and growth by condensation or deposition. The fundamental concept of the S89 scheme is that convergence of humidity within a grid-box contributes to an increase in total cloud water by condensation and deposition as well as an increase of the cloud fraction through moistening of the grid-box. Here we extend this idea to cold clouds and implement a cloud cover scheme which treats the microphysical structure of ice clouds consistently, requiring mixed-phase clouds to form via the liquid phase and allowing supersaturation in the cirrus regime.

## 2.1 Sub-grid cloud fraction

The original sub-grid cloud cover scheme of S89 does not consider cloud ice and assumes that a grid-box will be fully covered by the cloud if the water vapor mixing ratio $q_v$ surpasses the liquid water saturation mixing ratio $q_{sw}$, i.e. the relative humidity with respect to liquid water $s_w$ satisfies $s_w = q_v/q_{sw} = 1$. Cloud cover $b$ increases with increasing $s_w$ as:

$$b = 1 - \left( \frac{s_{max} - s_w}{s_{max} - s_{min}} \right)^{\frac{1}{2}}. \tag{1}$$

The parameter $s_{min}$ is the threshold relative humidity for sub-grid cloudiness below grid-box mean water saturation and $s_{max} = 1$. The assumptions underlying Eq. (1) are reasonable for synoptic-scale warm clouds. However, this scheme cannot be easily extended to cold clouds where a grid-box may or may not contain cloud ice in the mixed-phase regime, rendering the global use of $s_w$ for all temperatures inadequate. At the same time, relative humidity with respect to ice ($s_i$) can reach values higher than $140\,\%$ before ice nucleates from deliquesced aerosols (Koop et al., 2000), which is inconsistent with the use of $s_{max} = 1$ for ice clouds at cold temperatures. A few approaches exist to overcome either of these aspects. In the following we present the reasoning that lead to the development of the new method used here. For an overview, see Fig. 1.





As illustrated in the top panel of Fig. 1, Lohmann and Roeckner (1996) (the method used in the reference model) replaced $s_w$ in Eq. (1) by $s_i$ if a threshold amount of ice is exceeded within the grid-box. This approach has the draw-back that clouds artificially expand upon glaciation because of the lower saturation water vapor pressure over ice than over liquid water. A similar approach is taken in Morrison and Gettelman (2008) (and D18; illustrated by the middle panel in Fig. 1) where instead

of the discontinuous transition from liquid water to ice saturation, the saturation liquid and ice water vapor mixing ratios are interpolated between $q_{sw}$ and $q_{si}$ as a function of temperature in the mixed-phase regime. The draw-back here is, that this interpolated value is not directly relatable to a physical quantity as it does not arise from a valid solution of the Clausius-Clapeyron equation. Both of these schemes are not designed to allow for sub-grid clouds in the cirrus regime where $s_i$ needs to be well above 1 for ice nucleation to occur, i.e. all cirrus clouds formed by homogeneous freezing will occupy the entire

grid-box. Supersaturation with respect to cloud ice has been accounted for in the cloud cover scheme by Gettelman et al. (2010) where individual cloud fractions for liquid water and ice clouds are computed with a functional dependency of the cloud fraction $b(s)$ similar to Eq. (1) but relaxing the relative humidity for complete cloud coverage to $s_{max} = 1.1$ for ice clouds and treating the value of $s_{max}$ as a model parameter.

The cloud fraction parameterization used in our microphysics scheme is very similar to this last approach, but we do not

separate cloud cover for liquid and ice clouds. After all, in the mixed-phase regime we expect ice clouds to originate from liquid clouds (Hande and Hoose, 2017). Furthermore, the magnitude of super-saturation for complete cloud cover is rather arbitrary. The scheme presented here uses a limit which is consistent with the parameterization of ice nucleation based on the theory of Koop et al. (2000). This scheme is illustrated in the bottom panel of Fig. 1.

One can associate $s_{min}$ and $s_{max}$ in Eq. (1) with the minimal relative humidity below which clouds start to evaporate and

the maximal relative humidity which can be reached before the cloud covers the entire grid box. These two constraints provide some guidelines on how to extend the S89 scheme to cold clouds.

For clouds that form via liquid water, we assume that $s_{max}$ is reached at liquid water saturation, e.g. when liquid water is assumed to condense immediately, which is a reasonable assumption for stratiform clouds. At temperatures warmer than $-35\,^{\circ}\mathrm{C}$, any ice must form via liquid water, consistent with lidar observations and modeling studies (Ansmann et al., 2009;

Hoose et al., 2010). For colder temperatures, it has been shown (Koop et al., 2000) that ice can nucleate from deliquesced aerosols already below liquid water saturation.

The minimal relative humidity $s_{min}$ must be chosen such that ice does not sublimate above ice saturation. Cloud ice can sediment into and form within the mixed-phase regime. For this reason, and in order to retain sub-grid cloudiness for warm clouds, $s_{min} \leq 1$ must hold for all temperatures.

## 2.2 Growth by condensation and deposition

The water vapor available for condensation is linked to the cloud cover $b$ by:

$$Q = -b(\Delta q_v - \Delta q_{sw}) \tag{2}$$





where $\Delta q_v$ is the moisture convergence in the grid-box by the resolved transport and $\Delta q_{sw}$ is the change in liquid water saturation water vapor mixing ratio according to the Clausius-Clapeyron equation. Growth by deposition $A$ is computed as a function of the relative humidity with respect to ice $s_i$ (see e.g. D18):

$$A = \Delta t N_i \alpha_m f_v \frac{4\pi C (s_i - 1)}{F_k^i + F_d^i} \tag{3}$$

where $\alpha_m = 0.5$ is the probability of a water vapor molecule to successfully be incorporated into an ice crystal, $\Delta t$ is the model time step and $C$ is the mean ice particle capacitance integrated over the particle size distribution. The parameters $F_k^i$ and $F_d^i$ are thermodynamic parameters depending only on temperature.

The maximal amount of water that can deposit in a single time step is is given by:

$$Q_i = Q + q_l + (q_v - q_{si}) \tag{4}$$

where $q_l$ is the liquid water mixing ratio and $q_{si}$ is the saturation water vapor mixing ratio with respect to ice. Equation (4) implicitly represents the WBF process in mixed-phase clouds through the inclusion of $q_l$. The last term in Eq. (4) allows any water vapor which is supersaturated with respect to cloud ice to deposit onto ice crystals and thus requires to add $q_i$ to the water vapor mixing ratio for the computation of the relative humidity in Eq. (1) to prevent clouds from shrinking due to deposition. To account for ice clouds, we replace the relative humidity with respect to liquid water $s_w$ in Eq. (1) by a more general relative
humidity term $s$ which is defined below.

Combining the considerations for growth by condensation/deposition and the computation of cloud fraction, we obtain the key parameters for the computation of sub-grid cloudiness in Table 1. Two new parameters are introduced: the critical relative humidity with respect to liquid water for warm clouds $K$, which is equivalent to the parameterization for warm clouds used in the reference model and based on Xu and Krueger (1991), and the relative humidity required for nucleation of ice from
deliquesced aerosols $s_{koop}$ according to the activation theory of Koop et al. (2000) in the cirrus regime.

Two versions of the $s$ and $Q_i$ terms are given in Table 1 and represent two different methods to treat depositional growth in the mixed-phase regime. They bracket the main problem associated with cloud ice when diagnosing cloud cover from relative humidity. Cloud ice should be able to grow as long as water vapor is supersaturated with respect to ice. Since we are using the relative humidity with respect to liquid water (which is lower than the relative humidity with respect to ice) as maximal
relative humidity $s_{max}$ in the mixed-phase regime, ice growth below water saturation leads to a decrease in the cloud fraction $b$. Therefore we implement two versions of the new scheme. The first, default, method (subscripts 1) links cloud cover to the ice mass mixing ratio. It allows cloud ice to grow below liquid water saturation and adds cloud ice to water vapor for the computation of $s$ in Eq. (1) to avoid cloud shrinking by vapor deposition. However, this coupling makes the sedimentation sink of cloud ice also a sink for cloud fraction. We test the sensitivity of this coupling with a second method (subscripts 2) where
ice growth is inhibited below water saturation in the mixed-phase regime.



## 3 Model validation

We evaluate the new model against a series of satellite observations as well as the reference model and test sensitivities to parameterization choices. Namely, we extend the investigation of ice properties presented in D18 to the global setup of ECHAM-HAM and assess the behaviour of the new cloud cover parameterization. Furthermore, we use an updated version of

the cirrus parameterization, including heterogeneous ice nucleation of mineral dust at temperatures below $-35\,°C$ at low ice supersaturation based on Kärcher et al. (2006) (implementation courtesy Steffen Münch, personal communication).

Results from 5 different model configurations for 10 years from 2003 to 2012 are shown, one with the reference model (ECHAM6.3-HAM2.3; REF) and 4 with the new scheme. We assess the influence of rime properties in the full P3 scheme (4M) by comparison with a model configuration where the rime properties are set to zero ($q_{rim} = b_{rim} = 0$) (2M). For the

reasons presented below, we use the latter as the default configuration of the new model. With 2M as the starting configuration, we then investigate the effect of the artificial sedimentation-cloud cover feedback introduced by the new cloud cover scheme by limiting ice growth to liquid water saturation (LIM_ICE) and use an updated version of the cirrus cloud parameterization which includes heterogeneous ice nucleation on mineral dust below $-35\,°C$ (HET_CIR). The simulations and the employed tuning parameters are summarized in Table 2. The tuning parameters have been adjusted to meet the TOA energy flux constraints for

the simulations 2M, HET_CIR and REF.

### 3.1 Computational performance

The new model employs an adaptive time-stepping scheme to achieve numerical stability for the solution of the vertical advection equation for prognostic sedimentation of cloud ice. Sub-stepping the microphysics scheme comes at a cost, quantified by the CPU time column in Table 2. The numbers shown are computed from one year of simulation. Numbers vary by a few

percent between runs, illustrated by the differences between the 2M, LIM_ICE and HET_CIR simulations which are comparable in terms of model complexity. Compared to the reference model, the new model runs approximately $25\,\%$ slower in the 2M/LIM_ICE/HET_CIR setups and $40\,\%$ slower in the 4M setup. This difference is mainly due to the high fall-speeds of rimed particles in the 4M simulation, which require more sub-steps. Compared to this, the cost of advecting two additional tracers is small.

### 3.2 Comparison to the reference model

Two main differences between the new model and the reference model are immediately evident from Table 2. The single category does no longer require heuristic parameterizations for falling ice crystals and snow formation. At the same time, the new cloud cover parameterization allows to reduce the scaling factors, effectively bringing the parameterizations closer to their conceptual origin.

To illustrate the most prominent differences in the simulated cloud field, we compare cloud water contents (ice and liquid) as well as cloud cover in Fig. 2 of the new model in the 2M configuration to the reference model. For these quantities, the differences between the different configurations of the new model are small. It is evident, that the new model produces a lot



more ice than the reference model. A large part of the difference can be explained by the fact that the single category scheme includes cloud ice and snow while snow is a diagnostic quantity which is not included in the ice water content in the two-category scheme employed in the reference model. Of course there are other contributing factors like parameter tuning and the cloud cover parameterization which are hard to disentangle.

In the simulated cloud liquid water there is no significant structural difference; the 2M simulation has slightly higher values everywhere. As the warm phase is not directly affected by the changes in cloud ice and the cloud cover parameterization, the differences are due to different autoconversion tuning parameters ($\gamma_r$ in Table 2) and interactions with the ice phase.

The cloud cover differs significantly due to the new parameterization in the new model. Differences are most pronounced in the cirrus regime where a cloud now only covers the entire grid-box at the high supersaturation needed for homogeneous

nucleation of solution droplets. The comparison to the cloud cover climatology of the CALIPSO-GOCCP product (Cesana and Chepfer, 2013) in Fig. 3 reveals that the new scheme fits better to observations, despite an overall low bias. For this comparison we used the COSP simulator (Bodas-Salcedo et al., 2011).

## 3.3 Model tuning strategy

Model tuning has been conducted for the reference model (REF) and the two configurations of the new model 2M and

HET_CIR while the 4M and LIM_ICE configurations use the same tuning as 2M. A summary of global, annual mean quantities is shown in Table 3. The main target of the tuning process has been the global, annual mean shortwave (SW) and longwave (LW) fluxes at TOA as well as the sum of the two. With the tuning parameters summarized in Table 2 we have a direct handle on the net cloud radiative effect (CRE) at TOA, defined as the difference of all-sky and clear-sky radiative fluxes. To reach our SW and LW radiation targets we adjust the CRE. The fact that all the model simulations (including the reference model

with an substantially different microphysics scheme) find a CRE of roughly $-26\,\mathrm{W\,m^{-2}}$, which is more negative than any of the observational estimates, highlights that CRE is merely a result of tuning TOA radiative fluxes and not specific to the cloud microphysics scheme. The same is true for the hydrological cycle. Precipitation is largely governed by the rate of evaporation at the surface. As the sea surface temperature is prescribed an most evaporation occurs over the oceans, cloud microphysics and convection parameterizations must produce similar amounts of total surface precipitation.

The lower cloud cover of the 2M and 4M simulations as compared to the LIM_ICE simulation is a direct consequence of the coupling of cloud ice and cloud cover. There are two main contributors: 1) In the new scheme, the sedimentation sink of cloud ice is also a sink for cloud cover and 2) in the LIM_ICE configuration depositional growth of ice crystals is limited and therefore the removal of water vapor by ice sedimentation is weaker, leaving more humidity in the atmosphere which in turn leads to more clouds.

The 2M, 4M and HET_CIR configurations of the new model all allow for a more realistic zonal distribution of the LW CRE than the reference model, despite an overall low bias (Fig. 4). Without the overestimation of cirrus cloud cover, we can reduce the convective rain formation rate $\gamma_{cpr}$ to values closer to the pure ECHAM6 (without online aerosols from the HAM model and single moment cloud microphysics) model with a value of $2 \times 10^{-4}\,\mathrm{s^{-1}}$ (Mauritsen et al., 2012). This allows to



retain more water vapor in the updraft cores, thus enhancing the formation of tropical cirrus clouds in the outflow regions of convective anvils, associated with a strong LW CRE.

## 3.4 Cloud ice

We evaluate the representation of cloud ice in the new model against the data set compiled from CloudSat and CALIPSO
retrievals by Li et al. (2012) which include observations of cloud ice (CIWC) and total ice content (TIWC; including snow and ice in short-lived, convective anvils). The comparison is given in Fig. 5. With a single category, the cloud ice content cannot be unambiguously broken down into cloud ice and snow which is why TIWC is most representative. The reference model predicts cloud ice explicitly but represents snow as a mass flux. Therefore we compare the reference model to CIWC.

Comparing the different simulations of the new scheme, we see that the 2M and 4M simulations are very similar both in
terms of ice water content profiles (top row) and ice water path (TIWP and CIWP respectively; bottom). This suggests that the increased computational cost of the 4M configuration (see Table 2) and parameterization complexity (4 versus 2 prognostic ice moments) do not significantly improve the representation of cloud ice in the GCM.

As explained in Sect. 2, the new cloud cover parameterization couples the cloud fraction to the ice water content. The aggregation parameterization depends on the concentration of ice crystals $S_{agg} \propto N_i^2$, which in turn is computed from the
grid-box mean value as $N_i = \overline{N}_i b^{-1}$, i.e. $S_{agg} \propto b^{-2}$. We denote the parameterized source and sink terms by $S_{\bullet} = (\partial_t q_i)_{\bullet}$ i.e. the partial time derivative $\partial_t$ of cloud ice $q_i$ restricted to one process. This leads to an artificial feedback loop: Aggregation increases ice particle size, fall speed and thus the sedimentation sink. Because cloud ice is linked to cloud cover (see the definition of $s_1$ in Table 1 which replaces $s_w$ in Eq. (1) in the new cloud cover parameterization) the sedimentation of cloud ice decreases the cloud fraction $b$ which in turn increases the ice crystal aggregation rate $S_{agg}$ artificially. This feedback is the
main contributor to the overall slightly lower ice water contents in 2M simulation as compared to the LIM_ICE simulation, which does not have this feedback, and outweighs the effect from the additional humidity that is available for ice growth in the 2M simulation. We acknowledge the existence of this artificial feedback but do not consider it to be important enough to sacrifice the physical correctness of the method used in the 2M configuration.

The most prominent differences between our simulations and the observations are in the tropics. As can be seen from
Table 3, two thirds of the entire surface precipitation is produced by the convective scheme. Since convective precipitation is a diagnostic quantity which is not included in the total ice water content here, a direct comparison of modeled, stratiform IWC/IWP with the observations is limited. With that in mind, we find that the models underestimates the observed TIWP by roughly a factor of 3 in the tropics. This is true for both the new model and the reference model (which has to be compared to CIWP/CIWC). Slightly higher values for TIWP are produced by the LIM_ICE and HET_CIR simulations. While the higher
values in the LIM_ICE simulation are due to the cloud fraction - sedimentation feedback explained above, the higher values in the HET_CIR simulations are a result of the lower ice supersaturation needed for ice crystals to nucleate heterogeneously. This allows for a slightly more frequent formation of ice clouds at cold temperatures.

In the extra-tropics, the stratiform IWC is more representative of the observed TIWC/TIWP and we find slightly better agreement between the models and observations. The vertical profile of TIWC in the mid- and high latitudes is not reproduced





by the new model in any configuration, a feature shared with many CMIP5 models (Li et al., 2012), while the reference model performs rather well and only slightly underestimates the ice water path in mid-latitudes. As we will see in Sect. 5, the misrepresentation of the ice mass profile is likely due to unrealistic ice formation pathways in the new model.

On the basis of the evaluation above, we choose the 2M configuration as default for the new model for its computational
efficiency compared to the 4M configuration and physical correctness compared to the LIM_ICE configuration. Heterogeneous nucleation of ice crystals at low ice supersaturation in the cirrus regime (parameterized in the HET_CIR configuration) can be included optionally for future work with the new model but is not used for the analysis in Sects. 4 and 5 since its in part based on unpublished work.

### 3.5  Cloud liquid water

We compare the simulations with the new and the reference model to the Multisensor Advanced Climatology of Liquid Water Path (MAC-LWP) (Elsaesser et al., 2017), see Fig. 6. Microwave sensors cannot reliably distinguish precipitation from cloud water. Therefore we only evaluate regions with low precipitation, i.e. where the liquid water (LWP) and total water path (TWP) have similar magnitude. We use the threshold $LWP/TWP > 0.8$ suggested by Elsaesser et al. (2017). The differences in liquid water are very small for the different configurations of the new model, so we only look at the default configuration 2M.
We find an overall high bias in LWP in the extra-tropics for both models but a more pronounced effect in the new model. As discussed above, the TOA energy balance constraints in the new model require a lower tuning parameter for the autoconversion of cloud droplets to rain ($\gamma_r$) and thus thicker liquid clouds that reflect more SW radiation. We consider this the main reason for the higher liquid water path in the 2M as compared to the REF simulation, consistent with the findings of Lohmann and Neubauer (2018), as the warm phase parameterizations are the same in both models.

In the Southern Ocean, the overestimation of LWP translates into an overestimation of the SW CRE, evident from the top
right panel in Fig. 4. Note that the reference model with a smaller positive LWP bias overestimates SW CRE less.

### 3.6  Cloud phase partitioning

The CALIPSO-GOCCP product offers deeper insight into the cloud phase partitioning. Evaluating our models against this satellite retrieval reveals a significant shortcoming of the new as well as the reference model. We compute the frequency of
occurrence of the cloud phase ratio, defined as the fractional contribution of cloud ice to the total cloud water content, shown in Fig. 7. This metric is equivalent to Cesana et al. (2015) who present a comprehensive model inter-comparison of the phase ratio occurrence frequency. For the new model we only count cloudy regions, i.e. require $b > 0$, to exclude falling ice (snow) with a phase ratio of 1.

Both models capture the pronounced bimodality of the cloud phase distribution but significantly overestimate the frequency
of ice cloud occurrence at warm temperatures (and underestimate liquid cloud occurrence at cold temperatures). A whole family of models have a similar issue, as assessed by Cesana et al. (2015). This clearly suggests that there is a systematic error in the parameterization of cloud ice, the freezing process and ice growth in many models. To find the cause for this misrepresentation of the phase ratio, we present a novel approach to diagnose the cloud formation pathways in the next Section.





## 4  Quantifying the cloud formation pathways

The microphysical properties of a cloud are defined by the cloud formation processes, i.e. the cloud is simply the product of its formation history. Traditionally, model output reveals a snapshot of the simulated cloud field at any given time. Due to the finite storage capacity, aggregates of the model states are stored and information on the integration from process rates

is lost. Most common are temporal averages (e.g. monthly or yearly mean values) or vertical aggregates (e.g. burden, total cloud cover, TOA and surface radiative fluxes or precipitation). The aggregated cloud states are then compared to observations to infer information about the microphysical parameterizations leading to the cloud state. This last step provides significant uncertainty, is unnecessary and ultimately defeats the purpose of simulating any physical system: finding cause-and-effect between differential equations and observables.

In this section we tackle the problem the other way around. Instead of inferring the formation history based on the current cloud state, we make use of additional prognostic equations to store and quantify cloud formation pathways. This information can then be used to compute conditional probabilities to provide a sound cause-and-effect relation between the microphysical parameterizations and the resulting cloud state. This analysis makes direct inference from observables on cloud microphysical parameterizations tangible.

Our model parameterizes three fundamentally different source terms for cloud ice: Heterogeneous contact and immersion freezing of cloud droplets in the mixed-phase regime $S_{het-frz}$, homogeneous freezing of cloud droplets $S_{hom-frz}$ and nucleation of ice from deliquesced aerosols $S_{nuc}$. The latter two processes are restricted to temperatures below $-35\,°C$ while heterogeneous freezing dominates at warmer temperatures. Note that we focus on the mixed-phase regime and thus do not further separate heterogeneous from homogeneous ice nucleation in the cirrus regime, even though nucleation of cloud ice on

mineral dust is implemented in the HET_CIR configuration.

To distinguish ice that formed by either of the process rates above, we introduce two additional prognostic tracers to keep track of the formation history of the cloud mass at any given time in the simulation.

### 4.1  Mixed-phase heterogeneous freezing origin mass fraction

To separate cloud ice that formed in the mixed-phase temperature regime we introduce the heterogeneously nucleated ice mass

mixing ratio $q_{i,het}$ governed by the following equation:

$$\partial_t q_{i,het} = S_{het-frz} - v_m \partial_z q_{i,het} + F_{het}\left(S_{col} + S_{dep} - S_{sub}\right). \tag{5}$$

The growth of this tracer depends on the ice mass source and sink terms for collisions with cloud droplets $S_{col}$, vapor deposition $S_{dep}$, sublimation $S_{sub}$ and the fraction of cloud ice has already formed previously by heterogeneous freezing, defined as $F_{het} = q_{i,het}/q_i$. We abbreviate the partial derivative with respect to the vertical dimension $z$ (in m) by $\partial_z$. The tracer sediments

along with the total ice mass with the mass-weighted terminal velocity $v_m$ in $\mathrm{m\,s^{-1}}$. Convectively detrained ice is missing here due to a lack of an explicit, aerosol dependant freezing parameterization in the convection scheme. Water is only detrained as ice below temperatures of $-35\,°C$ where we assume that liquid water freezes homogeneously.



## 4.2 Liquid origin mass fraction

We further distinguish between cloud ice that forms via homogeneous freezing of cloud droplets and ice that nucleates in situ from deliquesced aerosol. This allows to disentangle in situ cirrus from liquid origin cirrus. To quantify the ice mass fraction initiated through freezing of liquid water, we implement a tracer $q_{i,liq-o}$ governed by the following equation:

$$\partial_t q_{i,liq-o} = S_{het-frz} + S_{ice-cv} + S_{hom-frz} - v_m \partial_z q_{i,liq-o} + F_{liq-o} \left( S_{col} + S_{dep} - S_{sub} \right). \tag{6}$$

We sum up all liquid to ice mass source terms, namely homogeneous freezing of cloud droplets $S_{hom-frz}$ and convective detrainment of cloud ice $S_{ice-cv}$ together with the processes defined above. The liquid origin mass fraction is given by $F_{liq-o} = q_{i,liq-o}/q_i$.

## 4.3 Cloud types based on the cloud formation history

The additional information on the cloud formation history from Eqs. (5) and (6) can be combined with the liquid fraction $F_{liq} = q_l/(q_i + q_l)$, cloud vertical thickness $\Delta p = p_{base} - p_{top}$, i.e. the pressure difference between cloud base $p_{base}$ and cloud top $p_{top}$, and cloud top temperature $T_{top}$ to build an inclusive set of cloud types. As is evident from Fig. 7, mixed-phase clouds are unstable due to the WBF process, leading to a strongly bimodal distribution of the frequency of phase ratio occurrence. This bimodality exists but is less pronounced for the ice mass source fractions $F_{het}$ and $F_{liq-o}$ shown in Fig. 8 together with the liquid fraction $F_{liq}$. Ice source mass fractions between zero and one arise from competing formation mechanisms and mixing of clouds of distinct sources.

We make use of the separation of cloud occurrence frequency in the 5-dimensional parameter space ($F_{liq}$, $\Delta p$, $F_{het}$, $F_{liq-o}$, $T_{top}$) to classify clouds into the types defined Table 4. The labels are based on the analysis of the temperature regime each cloud primarily occurs in and are discussed in more detail below.

The classification is computed online and thus allows to accumulate statistics per cloud type. Examples of such statistics are the cloud top temperature, liquid fraction and liquid origin fraction distributions per cloud type shown in Fig. 9. These distributions allow to identify physical properties of each cloud type and its associated formation pathway. Below we use them to justify the cloud type labels in Table 4.

Homogeneously nucleated clouds are separated into three classes. They are labeled *cirrus* if they are thinner than $500\,\mathrm{hPa}$, otherwise we refer to them as *thick*. This name stems from the fact that the vast majority of such clouds have cloud tops that are colder than $-35\,°\mathrm{C}$ as can be seen from the top panel in Fig. 9. We further differentiate between *in situ* cirrus, clouds that nucleated from deliquesced aerosols or formed heterogeneously below $-35\,°\mathrm{C}$ in the case of the HET_CIR simulation (not shown), and *liquid origin* cirrus, clouds that formed from homogeneous freezing of cloud droplets. While there is a clear peak for clouds that formed without any liquid water precursors ($F_{liq-o} = 0$), the peak for complete liquid origin clouds $F_{liq-o} = 1$ is less pronounced due to competing primary ice source terms (Fig. 9, bottom panel).

Heterogeneously nucleated clouds are separated into two classes. We find two distinctly different cloud types where ice formed predominantly from heterogeneous freezing of cloud droplets. As shown in Figs. 7 and 8, a truly mixed state with liquid





water and ice is unstable and thus rare. Therefore we divide heterogeneously nucleated clouds into those that are dominated by liquid water and ice respectively, see middle panel in Fig. 9. We label them both as *mixed-phase* (*liquid* and *ice dominated* respectively) since the cloud tops of such clouds are predominantly found in the mixed-phase temperature regime between $0\,°\mathrm{C}$ and $-35\,°\mathrm{C}$.

Liquid clouds are further separated into those with *cold* cloud tops ($T < 0\,°\mathrm{C}$) and *warm* cloud tops ($T > 0\,°\mathrm{C}$).

    In the following sections we will use the cloud types and associated formation pathways described here to gain insights into the simulated cloud fields. It is crucial to keep in mind that the labels provided for the cloud types here area based on formation pathways. This is in strong contrast to the traditional definitions based exclusively on the cloud state and can thus lead to results that seem counterintuitive at first. For example, cirrus clouds are commonly defined as clouds with temperatures

colder than $-35\,°\mathrm{C}$ (or more generally the temperature below which homogeneous freezing becomes efficient, depending on the model). Here we only require cirrus clouds to form by the processes that are typically found at temperatures below $-35\,°\mathrm{C}$ and subsequently track their evolution without imposing strict temperature constraints. This means however, that a cirrus cloud is still called cirrus, even if its constituent ice crystals fall far into the mixed-phase regime.

## 5   The ice formation pathways in ECHAM-HAM

With the additional prognostic equations to identify cloud formation pathways and the cloud types derived thereof, we are able find the microphysical cause for the macrophysical cloud state simulated by the new model. Identifying unrealistic cloud formation pathways allows to directly target faulty microphysical parameterizations or define tuning targets.

### 5.1   Relative cloud type frequencies

We classify the clouds online according to Table 4 to generate a global climatology of the relative contributions of each

formation pathway to the 3D cloud volume (defined as the air mass covered by the cloud) in Fig. 10. From this pie chart it is evident that the majority of the cloud population consists of homogeneously nucleated ice clouds (in situ cirrus and thick clouds) and liquid clouds. The next smaller class are the liquid origin cirrus clouds, making up almost $9\,\%$ of the cloud population or almost one fourth of all cirrus clouds. Mixed-phase clouds only contribute roughly $7\,\%$ with the fraction of such clouds that are dominated by ice water being smaller than those being dominated by liquid water.

We also show a more detailed map of cloud type occurrence frequency in Fig. 11. This view sheds more light on where each formation pathway dominates. Liquid clouds with warm cloud tops dominate in the tropics while their cold top counterparts are primarily found in mid-latitudes. Due to abundant deep convection in the tropics, large amounts of liquid water are transported to high altitudes where homogeneous freezing of cloud droplets sets in, leading to a frequent occurrence of liquid origin cirrus clouds. Heterogeneous freezing only affects clouds at low altitudes where there is no competition with homogeneous

nucleation. Seemingly inexistent are mixed-phase clouds where heterogeneous freezing of cloud droplets causes glaciation of the host liquid cloud.



## 5.2 The mixed-phase ice overestimation

The absolute frequency of occurrence of cloud types shown in the top panel in Fig. 12 reveals the pathways responsible for the global mean overestimation of the ice fraction in the mixed-phase regime as compared to the CALIPSO-GOCCP product (discussed above and shown in Fig. 7). The most frequent cloud types simulated in the mixed-phase regime are thick clouds,

in situ cirrus, liquid origin cirrus and only then followed by the two heterogeneously nucleated cloud types. The fact that we find homogeneously nucleated clouds with a vertical extent of less than $500\,\mathrm{hPa}$ (i.e. cirrus clouds) down to temperatures of $0\,°\mathrm{C}$ is in part due to the choice of the threshold value to separate cirrus clouds from thick clouds but does not affect the main conclusion: The dominant source term for cloud ice in the mixed-phase temperature regime is homogeneous freezing of deliquesced aerosols taking place at temperatures below $-35\,°\mathrm{C}$.

Analogously to the prognostic tracers introduced in Sect. 4 and formally defined in Eq. (A1), we trace the state of water from which ice forms. The frozen liquid ice mass tracer accumulates the mass of liquid water that has been converted to ice by freezing, riming or the WBF process and allows to compute the frozen liquid fraction as $F_{liq-f} = q_{i,liq-f}/q_i$. For the majority of homogeneously nucleated clouds, the frozen liquid fraction is very small ($< 10\,\%$) from which we conclude that ice predominantly forms directly from deposition of water vapor (excluding the WBF process) and thus without a liquid water

precursor, see the bottom panel in Fig. 12. We conclude that homogeneously nucleated ice not only dominates the frequency of occurrence of ice in the mixed-phase regime but also inhibits the formation of liquid clouds and thus heterogeneously nucleating ice clouds by quickly taking up any liquid water encountered on the fall-trajectory of the ice crystals, making vapor deposition the main ice growth term.

    This offers two explanations for the differences between the modeled phase ratio and the CALIPSO-GOCCP product. Either

the model produces too many thick clouds or the satellite is not able to detect the vertical extent of such clouds due to attenuation and therefore underestimates the frequency of ice occurrence at warmer temperatures.

    We can think of four mechanisms that lead to an overestimation of thick clouds in the model, listed in descending likelihood according to our judgment:

**Cloud overlap** Even though we take horizontal sub-grid clouds into account, we assume that the layers are maximally over-

25         lapped for the computation of sedimentation. If thick clouds consist of two vertically adjacent but not horizontally over-
        lapping clouds we underestimate sublimation in cloud-free areas and overestimate vertical transport within the clouds.

**Vertical resolution** Low vertical resolution does not resolve thin cloud layers. The vertical grid-spacing of more than $500\,\mathrm{m}$
        above $600\,\mathrm{hPa}$ is not adequate to resolve thin cirrus cloud layers which are only several $100\,\mathrm{m}$ thick. Our model thus
        underestimates cloud-free, sub-saturated areas in this altitude region and it may overestimate vertical mixing.

**Ice sedimentation** The new scheme vastly improves the physical realism of the sedimentation process (see D18). If sedimenta-
        tion is overestimated, it must be due to unrealistic microphysical properties of the crystals, produced by the homogeneous
        nucleation and freezing parameterizations. This does not seem very likely because we see a very similar behavior for





the HET_CIR configuration of the new model with different cirrus microphysics (not shown). This suggests that the frequency of occurrence of thick clouds is not very sensitive to the way cirrus clouds are parameterized.

**Single category**  A single category has the fundamental problem, that it cannot represent more than one particle population. Advective mixing of two or multiple ice particle populations leads to a combined, average population. It has been shown in D18 by an idealized seeder-feeder simulation that this artificial mechanism can trap large particles sedimenting into clouds with many small particles because the combined population has a significantly smaller average fall speed than that of the seeder population, thus overestimating the seeder-feeder process. However, as has been shown above (bottom panel in Fig. 12), mixed-phase processes have a negligible impact on homogeneously nucleated clouds. This is further verified by sensitivity simulations where mixed-phase freezing, riming and the WBF process are turned off (not shown). This mechanism can thus be ruled out.

This analysis reveals unequivocally that the overestimation of cloud ice in the mixed-phase regime and the underestimation of super-cooled liquid water cannot be associated with heterogeneous freezing and ice growth in the mixed-phase regime itself but is due to homogenous freezing and subsequent sedimentation of ice at colder temperatures.

## 6   Conclusions

We presented a new cloud microphysics scheme in the ECHAM6-HAM2 GCM. The main improvement over its predecessor is the consistent description of cloud ice using a single, prognostic category. Thus, it does not rely on poorly constrained conversion parameterizations between in-cloud ice and precipitating snow categories. We introduced a new approach to extend the sub-grid cloud cover scheme of Sundqvist et al. (1989) to ice clouds. This scheme does no longer have the positive cloud cover bias at temperatures below $-35\,°C$ from the reference model and therefore allows for smaller, arguably more reasonable tuning parameters.

We assessed cloud formation pathways quantitatively by introducing additional prognostic equations for the heterogeneously formed and liquid origin ice mass mixing ratios. We found that in our model the majority of cloud ice forms below $-35\,°C$ by either homogeneous freezing of cloud droplets or homogeneous nucleation of deliquesced aerosols. Only about $7\,\%$ of the cloud volume is dominated by heterogeneous freezing in mixed-phase temperatures, making homogeneous freezing (and heterogeneous nucleation of cloud ice on mineral dust at temperatures colder than $-35\,°C$ in the HET_CIR configuration, not shown) the main source for cloud ice even at temperatures warmer than $-35\,°C$. The Lagrangian perspective on the modeled cloud field provided by the formation pathway analysis allowed to distinguish in situ and liquid origin cirrus clouds. We found that roughly one forth of all cirrus clouds form from homogeneous freezing of cloud droplets.

Furthermore, cloud formation pathways provide the causal link between the microphysical parameterizations and the resulting cloud field. With an example for the cloud phase ratio, we showed that this link can be used as a powerful tool to use satellite products to pinpoint unrealistic parameterizations. A comparison with the CALIPSO-GOCCP product revealed that our model underestimates the frequency of occurrence of supercooled liquid water (and overestimates the frequency of cloud





ice occurrence) in the mixed-phase regime. Through the analysis of cloud formation pathways we could identify the responsible microphysical parameterizations for this discrepancy. Interestingly, we found that it is not the processes taking place in the mixed-phase regime itself (WBF process, heterogeneous freezing of cloud droplets and riming) which are responsible for this overestimation but rather a combination of ice formation processes at temperatures colder than $-35\,°C$ and subsequent

sedimentation. This suggests that model improvements regarding the cloud phase ratio should relax the maximal cloud overlap assumption for sedimentation.

*Code availability.*   The model code is available as part of the ECHAM6-HAMMOZ chemistry climate model through the HAMMOZ distribution web-page https://redmine.hammoz.ethz.ch/projects/hammoz.

## Appendix A: Additional prognostic equations to diagnose cloud formation

### A1    Frozen liquid mass fraction

We do not only diagnose the phase which initiated ice formation to separate liquid origin from vapor origin ice but also keep track of the total amount of liquid water that has been converted to ice with a separate tracer $q_{i,liq-f}$, the mass mixing ratio for frozen liquid water defined as:

$$\partial_t q_{i,liq-f} = S_{het-frz} + S_{ice-cv} + S_{hom-frz} + S_{col} - v_m \partial_z q_{i,liq-o} - F_{liq-f} S_{sub}. \tag{A1}$$

Analogous to the tracers defined in Section 4 we define the frozen liquid fraction as $F_{liq-f} = q_{i,liq-f}/q_i$. The difference to the liquid origin mass fraction is that deposition is not taken into account.

*Competing interests.*   The authors declare that they have no conflict of interest.

*Acknowledgements.*   This project has been funded by the Swiss National Science Foundation (project number 200021_160177). The ECHAM-HAMMOZ model is developed by a consortium composed of ETH Zurich, Max Planck Institut für Meteorologie, Forschungszentrum Jülich,

University of Oxford, the Finnish Meteorological Institute and the Leibniz Institute for Tropospheric Research, and managed by the Center for Climate Systems Modeling (C2SM) at ETH Zurich. Special thanks go to Sylvaine Ferrachat for technical support regarding the model. The computing time for this work was supported by a grant from the Swiss National Supercomputing Center (CSCS) under project ID s652 and from ETH Zurich. We thank Steffen Münch for providing his implementation of a more advanced cirrus parameterizations for the sensitivity tests presented here.





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





**Table 1.** The parameters involved in the cloud cover scheme, Eqs. (1) and (4). The rows show the saturation ratios as a function of the temperature regime $T$. We discuss two different choices for $s$ and $Q_i$ in the text, $s_1/Q_{i,1}$ and $s_2/Q_{i,2}$. We use a linear weighting function $w(T) = (T - 273.15)/(238.15 - 273.15)$ for $T$ in K.

|  | $T < -35\,^{\circ}\mathrm{C}$ | $-35\,^{\circ}\mathrm{C} < T < 0\,^{\circ}\mathrm{C}$ | $0\,^{\circ}\mathrm{C} < T$ |
|---|---|---|---|
| $s_1$ | $(q_v + q_i)/q_{si}$ | $(q_v + q_i)/q_{si}$ | $q_v/q_{sw}$ |
| $Q_{i,1}$ | $q_v - q_{si}$ | $Q + q_l + (q_v - q_{si})$ | |
| $s_2$ | $(q_v + q_i)/q_{si}$ | $q_v/q_{si}$ | $q_v/q_{sw}$ |
| $Q_{i,2}$ | $q_v - q_{si}$ | $Q + q_l$ | |
| $s_{max}$ | $s_{koop}$ | $q_{si}/q_{sw}$ | $1$ |
| $s_{min}$ | $1$ | $w(T) + (1 - w(T))K$ | $K$ |

Tan, I., Storelvmo, T., and Zelinka, M. D.: Observational constraints on mixed-phase clouds imply higher climate sensitivity, Science, 352, 224–227, doi:10.1126/science.aad5300, 2016.

Terai, C. R., Klein, S. A., and Zelinka, M. D.: Constraining the low-cloud optical depth feedback at middle and high latitudes using satellite observations, J. geophys. res-atmos., 121, 9696–9716, doi:10.1002/2016JD025233, 2016.

5 Tsushima, Y., Ringer, M. A., Koshiro, T., Kawai, H., Roehrig, R., Cole, J., Watanabe, M., Yokohata, T., Bodas-Salcedo, A., Williams, K. D., and Webb, M. J.: Robustness, uncertainties, and emergent constraints in the radiative responses of stratocumulus cloud regimes to future warming, Clim. dyn., 46, 3025–3039, doi:10.1007/s00382-015-2750-7, https://doi.org/10.1007/s00382-015-2750-7, 2016.

Welti, A., Kanji, Z. A., Lueoend, F., Stetzer, O., and Lohmann, U.: Exploring the Mechanisms of Ice Nucleation on Kaolinite: From Deposition Nucleation to Condensation Freezing, J. atmos. sci., 71, 16–36, doi:10.1175/JAS-D-12-0252.1, 2014.

10 Wernli, H., Boettcher, M., Joos, H., Miltenberger, A. K., and Spichtinger, P.: A trajectory-based classification of ERA-Interim ice clouds in the region of the North Atlantic storm track, Geophysical Research Letters, 43, 6657–6664, doi:10.1002/2016GL068922, 2016.

Williams, K. D. and Tselioudis, G.: GCM intercomparison of global cloud regimes: present-day evaluation and climate change response, Clim. dyn., 29, 231–250, doi:10.1007/s00382-007-0232-2, https://doi.org/10.1007/s00382-007-0232-2, 2007.

Williams, K. D. and Webb, M. J.: A quantitative performance assessment of cloud regimes in climate models, Clim. dyn., 33, 141–157, 15 doi:10.1007/s00382-008-0443-1, https://doi.org/10.1007/s00382-008-0443-1, 2009.

Xi, Z., Yanluan, L., Yiran, P., Bin, W., Hugh, M., and Andrew, G.: A single ice approach using varying ice particle properties in global climate model microphysics, J. adv. model. Earth syst., 9, 2138–2157, doi:10.1002/2017MS000952, 2017.

Xu, K.-M. and Krueger, S. K.: Evaluation of Cloudiness Parameterizations Using a Cumulus Ensemble Model, Monthly Weather Review, 119, 342–367, doi:10.1175/1520-0493(1991)119<0342:EOCPUA>2.0.CO;2, 1991.

20 Zelinka, M. D., Klein, S. A., Taylor, K. E., Andrews, T., Webb, M. J., Gregory, J. M., and Forster, P. M.: Contributions of Different Cloud Types to Feedbacks and Rapid Adjustments in CMIP5, J. climate, 26, 5007–5027, doi:10.1175/JCLI-D-12-00555.1, 2013.




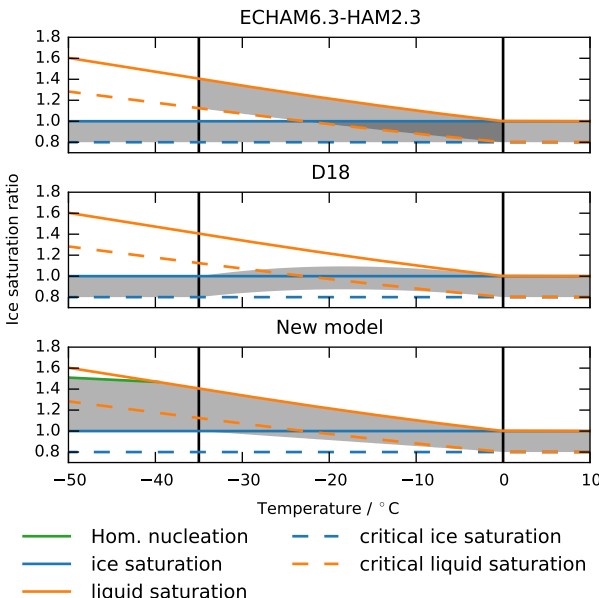

**Figure 1.** Regions where clouds can form (grey shaded areas) as a function of temperature and super saturation with respect to ice. The colored lines show liquid water and ice saturation ratios $s_{w/i}$ (solid lines) and the critical saturation ratios required for first cloud formation (dashed; $s_{w/i}K$). We choose a constant $K = 0.8$ for this illustration. The top panel shows the sub-grid cloud scheme of the reference model ECHAM6.3-HAM2.3 which switches between liquid and ice saturation in the mixed-phase regime, the middle panel shows the mixed-saturation scheme of D18 and the bottom panel shows the scheme of the new model.



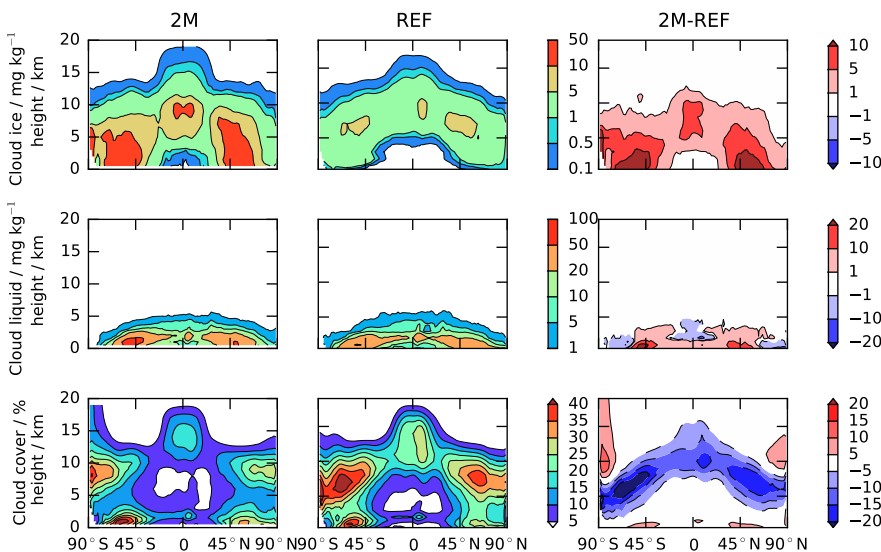

**Figure 2.** 10-year, zonal mean cloud ice water content (top row), cloud liquid water content (middle row) and cloud cover (bottom row) for the new model (2M configuration; left), the reference model (REF; middle) and differences between the two schemes (right).

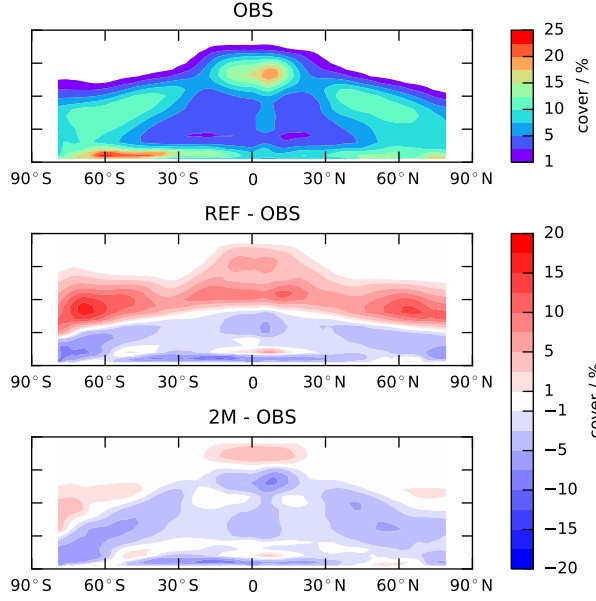

**Figure 3.** Comparison of the simulated cloud fraction for simulations with the 2M configuration of the new model and the reference model (REF) for 10 years (2003-2012) and the CALIPSO-GOCCP satellite product (7 years, 2008-2014). Model output is computed using the COSP-simulator.





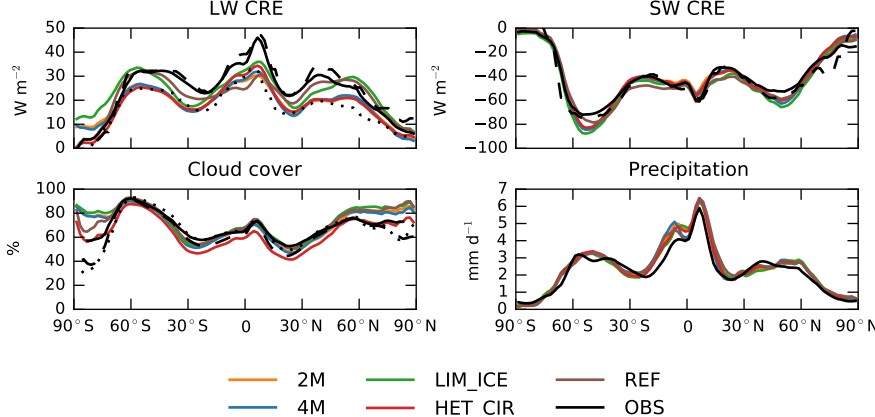

**Figure 4.** 10-year, zonal mean values for long- (LW CRE) and shortwave cloud radiative effects (SW CRE) (top row) and total cloud cover and precipitation (bottom row). The colored lines show simulations of different configurations of the new model in Table 2. Black lines show satellite products. For CRE we show data from CERES (solid, Loeb et al. (2018)), ERBE (dashed) and TOVS (LW only; dotted; Susskind et al. (1997)). Total cloud cover is shown for CALIPSO-GOCCP (solid; Chepfer et al. (2010)), ESA Cloud CCI project (AVHRR-PM) (dashed, Stengel et al. (2017)) and MODIS (dotted; Platnick et al. (2015, 2017)). Precipitation is from the global precipitation climatology product (GPCP) (solid; Adler et al. (2018))

**Table 2.** Description of the model configurations shown in this paper. Tuning parameters are as follows: $\gamma_r$ is the scaling factor for warm rain formation, $f_{fall}$ is a scaling factor for ice sedimentation speed, $\gamma_s$ is a scaling parameter for snow formation, $e_{ii}$ is the collision efficiency of ice crystals and $\gamma_{cpr}$ is the conversion rate from cloud water and ice to precipitation in the convection scheme. Dashes '−' denote that those parameters are no longer needed in the new scheme. The last column shows the CPU time for 1 year of simulation relative to the reference model.

| Simulation | Description | $\gamma_r$ | $f_{fall}$ | $\gamma_s$ | $e_{ii}$ | $\gamma_{cpr}\ (s^{-1})$ | CPU time |
|---|---|---|---|---|---|---|---|
| REF | Reference version: ECHAM6.3-HAM2.3 | 10.6 | 3 | 900 | $0.09e^{T_c}$ | $9 \times 10^{-4}$ | − |
| 2M | Single category, 2 prognostic ice moments | 7 | − | − | 0.5 | $2.5 \times 10^{-4}$ | +29 % |
| 4M | Single category, 4 prognostic ice moments | 7 | − | − | 0.5 | $2.5 \times 10^{-4}$ | +41 % |
| QSW | As 2M but deposition below $q_v = q_{s,w}$ is shut down for $T > -35\,°C$. | 7 | − | − | 0.5 | $2.5 \times 10^{-4}$ | +24 % |
| HCI | As 2M but with het. nucleation in cirrus regime | 9.5 | − | − | 0.5 | $1.5 \times 10^{-4}$ | +24 % |





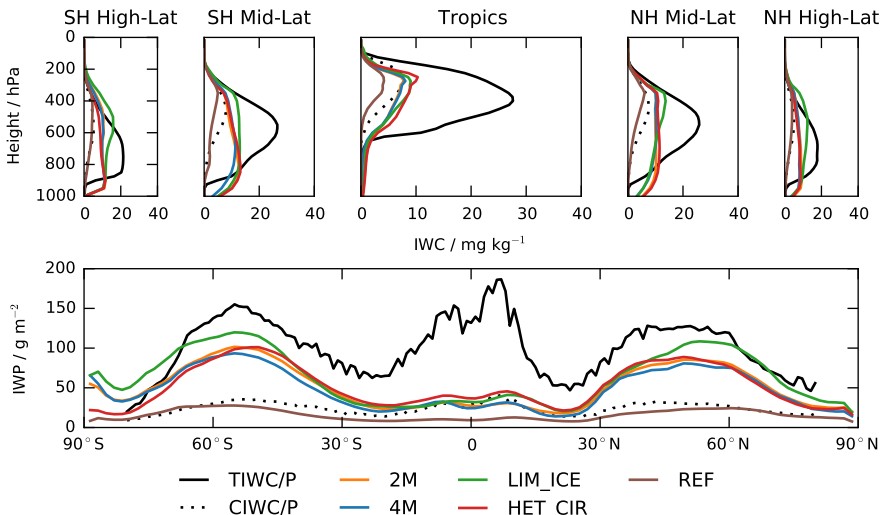

**Figure 5.** Ice water content and ice water path compiled from observations by Li et al. (2012) and different versions of ECHAM6-HAM2 (10 year, zonal averages) with the new microphysics scheme and the reference model. The top row shows vertical profiles for ice water content, the panel on the bottom shows ice water path. The observations in black include total ice water content/path (TIWC/P; solid lines) and cloud ice water content/path (CIWC/P; dotted lines). Colors show total (stratiform) ice water content from the new model and cloud ice water content from the reference model. The profiles are averaged over different latitude bands: SH High-Lat (80°S to 60°), SH Mid-Lat (60°S to 30°S), Tropics (30°S to 30°N), NH Mid-Lat (30°N to 60°N) and NH High-Lat (60°N to 80°N).



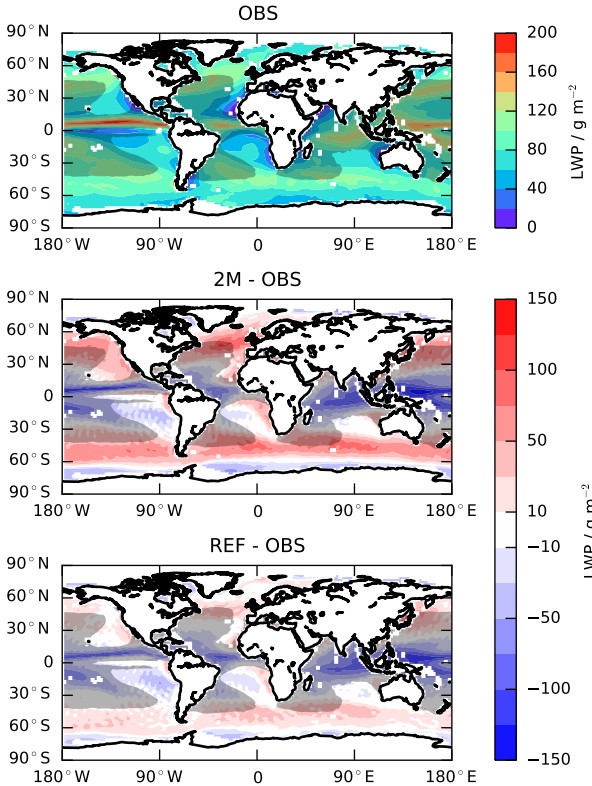

**Figure 6.** LWP as observed by the Multisensor Advanced Climatology of LWP (MAC-LWP), climatology from 2003 to 2012. The top panel shows the observations, the middle shows the difference between the new model (2M) and the observations and the bottom panel shows the difference between the reference model and the observations. The models are both run for 10 years from 2003 to 2012. The gray shaded areas show regions where the liquid water path is dominated by precipitation ($LWP/TWP < 0.8$), i.e. where there is no reliable estimate for in-cloud liquid water path.





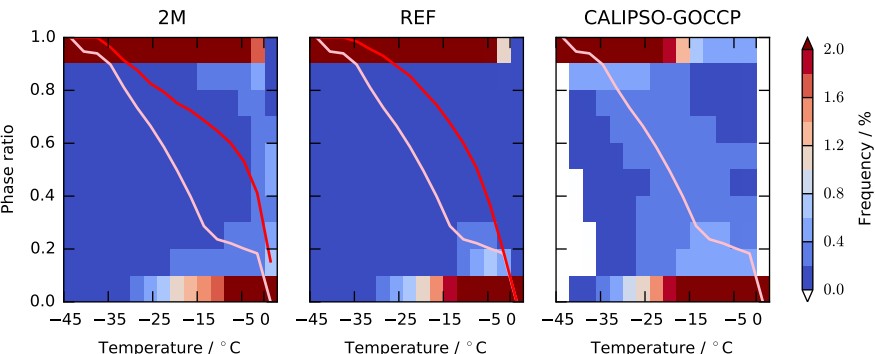

**Figure 7.** Frequency of cloud occurrence per temperature and phase ratio for histogram bin widths of 3 K and 0.1 respectively. The average ice fraction per temperature bin is shown by the red line for the new model (2M; left) and the reference model (REF; middle). The pink line appears in every subplot and shows the ice fraction from the CALIPSO-GOCCP (Cesana et al., 2015) product. Both the models and observations are accumulated over night only. The models contain data from 10 years simulation from 2003 to 2012. The satellite data is accumulated over 7 years from 2007 to 2013.

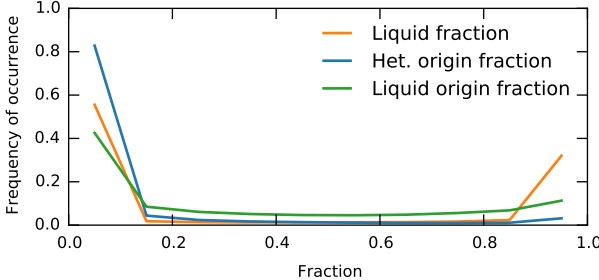

**Figure 8.** Frequency of cloud occurrence per ice source fraction for the liquid origin fraction $F_{liq}$ and the different ice source fractions $F_{het}$ and $F_{liq-o}$. The histogram bin width is 0.1. The lines are aligned with bin centers. Only gridboxes are sampled where all the fractions are well-defined, i.e. only cloudy gridboxes for $F_{liq}$ and only gridboxes containing ice for $F_{het}$ and $F_{liq-o}$. Data is sampled every time step for 10 years of simulation.





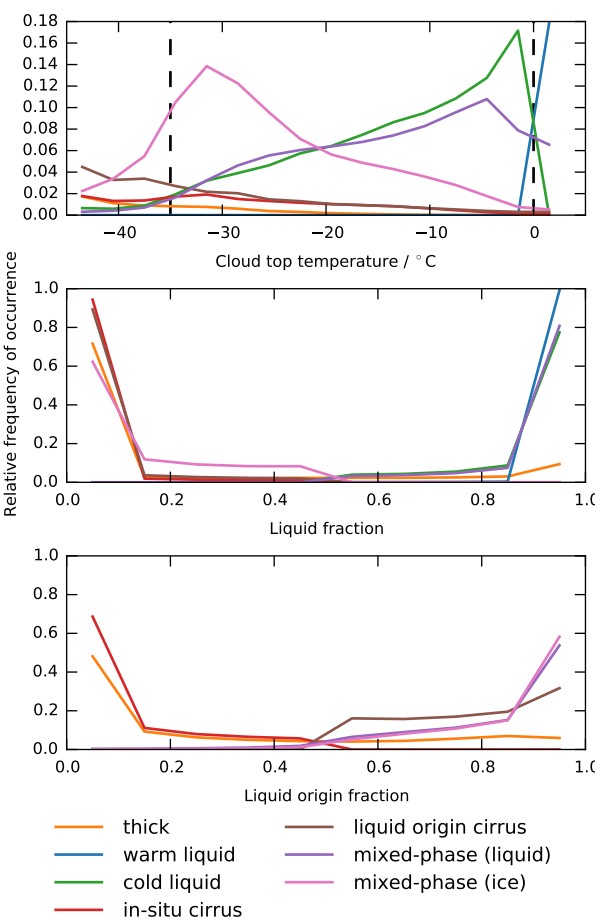

**Figure 9.** Relative frequency of cloud occurrence (normalized by the cloud type occurrence frequency). This corresponds to conditional probability histograms, $P(X_\bullet \in Bin | X \in Type_i)$ for the model state $X$ with components $X_\bullet$: cloud top temperature (top panel), liquid fraction (middle panel) and liquid origin fraction (bottom panel) for each cloud type $i$. Bin widths are $3\,\text{K}$ for temperature and $0.1$ for the dimensionless fractions respectively, lines are aligned with bin centers. Data is sampled every time step for 10 years of simulation.





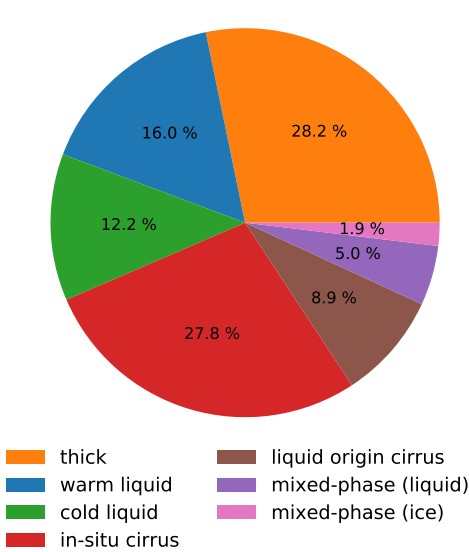

**Figure 10.** Global relative contribution to the 3D cloud volume for the cloud types defined in Table 4 based on their formation pathways. Data is sampled every time step for 10 years of simulation.





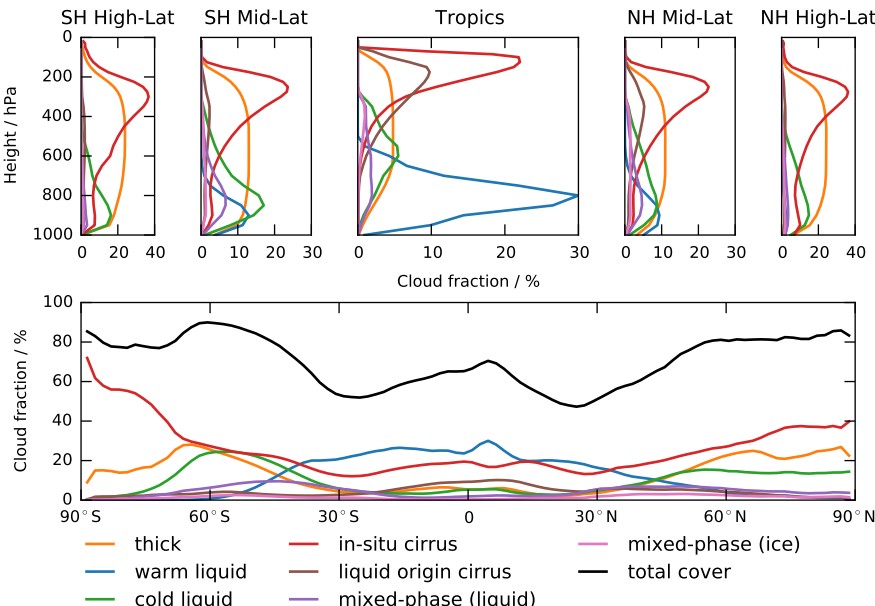

**Figure 11.** Same as in Fig. 5 but for the absolute frequency of occurrence profile (top panel) and relative contribution to the total cloud cover (bottom panel). The relative contribution is computed as the fraction of cloud volume occupied per column and cloud type. Data is sampled every time step for 10 years of simulation.





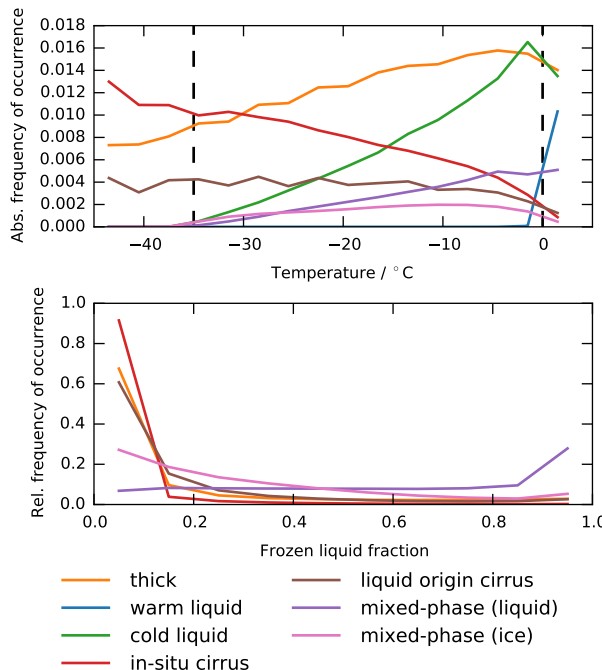

**Figure 12.** Frequency of occurrence for grid-box temperature (top) and frozen liquid fraction (bottom). For the temperature component $X_T$ of the model state $\boldsymbol{X}$ we show the absolute frequency of occurrence $P(X_T \in Bin | \boldsymbol{X} \in Type_i) \cdot P(\boldsymbol{X} \in Type_i | X_b > 0)$. We only sample cloudy grid-boxes, i.e. those with a cloud fraction component $X_b > 0$. For the frozen liquid fraction component $X_{F_{liq-f}}$ we show the relative frequency of occurrence $P(X_{F_{liq-f}} \in Bin | \boldsymbol{X} \in Type_i)$. The latter is the accumulated ice mass fraction that formed through any liquid water to ice conversion process (including the WBF process). Note that this is not equal to the liquid origin mass that quantifies the origin of cloud ice. Bin widths are $3\,\mathrm{K}$ and $0.1$ respectively, lines are aligned with bin centers. Data is sampled every time step for 10 years of simulation.


**Table 3.** 10-year, global average values for a selection of key microphysical parameters for the model configurations presented in Table 2. Observational data is used from the compilation of Lohmann and Neubauer (2018).

| Simulation | OBS | 2M | 4M | LIM_ICE | HET_CIR | REF |
|---|---|---|---|---|---|---|
| LWP, g m$^{-2}$ | 81.4 (30 to 90) | 76.1 | 80.9 | 77.3 | 70.9 | 64.9 |
| TIWP, g m$^{-2}$ | 100.8 | 48.7 | 44.9 | 57.4 | 52.9 | - |
| CIWP, g m$^{-2}$ | 24.2 | - | - | - | - | 14.8 |
| $N_c$, $10^{10}$ m$^{-2}$ | - | 4.67 | 4.69 | 4.57 | 4.61 | 3.16 |
| $N_i$, $10^{10}$ m$^{-2}$ | - | 0.11 | 0.10 | 0.14 | 0.07 | 0.08 |
| CC, % | 68±5 | 65.9 | 65.9 | 68.3 | 59.9 | 68.3 |
| Q, kg m$^{-2}$ | 25.7 | 27.39 | 27.40 | 28.06 | 27.53 | 26.41 |
| $P_{tot}$, mm d$^{-1}$ | 2.7±0.2 | 2.9 | 3.0 | 2.9 | 2.9 | 3.0 |
| $P_{strat}$, mm d$^{-1}$ | - | 1.0 | 1.0 | 1.0 | 1.1 | 1.0 |
| $P_{cnv}$, mm d$^{-1}$ | - | 1.9 | 1.9 | 1.9 | 1.8 | 2.0 |
| SW CRE, W m$^{-2}$ | -47.3 (-44 to -53.3) | -47.3 | -47.9 | -50.6 | -47.0 | -50.1 |
| LW CRE, W m$^{-2}$ | 26.2 (22 to 33.5) | 20.9 | 21.0 | 25.3 | 21.2 | 24.3 |
| Net CRE, W m$^{-2}$ | -21.1 (-17.1 to 22.8) | -26.4 | -26.9 | -25.3 | -25.8 | -25.8 |
| TOA LW, W m$^{-2}$ | -(237 to 241) | -238.6 | -238.8 | -234.7 | -238.9 | -237.8 |
| TOA SW, W m$^{-2}$ | 238 to 244 | 239.3 | 238.9 | 236.5 | 239.1 | 238.1 |
| $\Delta$ TOA, W m$^{-2}$ | | 0.7 | 0.1 | 1.9 | 0.2 | 0.3 |





**Table 4.** Definition of cloud types. We separate clouds by being larger or smaller than a given threshold value (symbolized by $>$ and $<$ respectively) for each of the four predictors: The heterogeneous freezing origin fraction $F_{het}$, cloud top to bottom pressure difference $\Delta p$, cloud liquid fraction $F_{liq}$ and liquid-origin fraction $F_{liq-o}$. The threshold value for the dimensionless fractions is 0.5, for the pressure difference it is $500\,\text{hPa}$ and for the cloud top temperature $0\,°\text{C}$. If a predictor is not used for a certain class, it is symbolized by a '$-$' sign.

| Label | $F_{het}$ | $\Delta p$ | $F_{liq}$ | $F_{liq-o}$ | $T_{top}$ |
|---|---|---|---|---|---|
| Thick clouds, homogeneous origin | $<$ | $>$ | $-$ | $-$ | $-$ |
| warm liquid clouds | $<$ | $<$ | $>$ | $-$ | $>$ |
| cold liquid clouds | $<$ | $<$ | $>$ | $-$ | $<$ |
| Cirrus clouds, in situ | $<$ | $<$ | $<$ | $<$ | $-$ |
| Cirrus clouds, liquid origin | $<$ | $<$ | $<$ | $>$ | $-$ |
| Mixed-phase, liquid dominated | $>$ | $-$ | $>$ | $-$ | $-$ |
| Mixed-phase, ice dominated | $>$ | $-$ | $<$ | $-$ | $-$ |