# Peer review of "Elucidating ice formation pathways in the aerosol-climate model ECHAM6-HAM2"

_Atmospheric Chemistry and Physics, 2018_

## Referee Comment (RC1) · Anonymous Referee #1 · 6 Aug 2018

Decision: The authors study the impact of a new microphysics scheme in a GCM on the representation of cloud and cloud phase partition in the present-day climate. They introduce the differences between their new microphysics scheme and the default one along with some additional hybrid versions of their new scheme. Then, they evaluate the cloud and radiation fields of their new model against a large set of observations. Finally, after identifying a cold bias in the liquid and ice partitioning between 0 and -35ËŽC (the mixed-phase temperature range), they investigate the reasons of this bias using a Lagrangian method, which allows one to tracing the origin of the overestimated ice clouds. It appears that these extra-ice clouds within the mixed-phase temperatures mostly comes from ice cloud formed at temperatures below -35ËŽC, i.e., out of the mixed-phase temperature range.

[Figure]

This paper is within the scope of the journal and generally well written, clearly presented and easy to understand. The topic is of particular interest as it investigates a persisting problem in state-of-the-art GCMs, i.e., the lack (excess) of liquid (ice) clouds. The authors use an original method to tackle this problem and add up interesting results to previous literature. However, the model evaluation part suffers from important flaws (further described below), which should be addressed before final publication. In addition, I believe extra-analysis could be accomplished to strengthen the conclusion of the paper and I provide some way of doing it below. Therefore, I would recommend a major revision before considering the paper for publication.

Main comments: In the model evaluation section, the authors do not describe the dataset at all and do not explain why they chose these specific datasets to evaluate their model. For example, three datasets are used for the fluxes and no reason whatsoever is given to justify this choice. The authors must define/introduce the datasets, even briefly, and explain why they use these. Also, the observed interannual STD can be used as an uncertainty estimates when nothing else is available. In addition, their model evaluation is more of a qualitative comparison than a quantitative one although it is possible to quantify the bias more precisely (see specific comments for further details). It also looks like (it is not specified in the manuscript) they didn't make a consistent comparison between the CALIPSO-GOCCP cloud phase dataset and their model outputs, i.e., they didn't use the simulator for the cloud phase diagnostics, which make the results difficult to interpret. In the second part of the manuscript, I feel like reducing the number of categories in the final part of the study would help to better determine the origin of the ice bias. I'd like the author to either do that or better explain why they chose these categories and what is the added value of making this choice. Finally, the authors could easily check at least one of the mechanism that supposedly lead to the overestimation of the ice cloud occurrences (the overlap assumption).

Specific comments (Introduction): The goal of the paper is a little bit confused and not clearly stated. Also, it is not clear to me how "As has been eluded to above, the

formation history of a cloud plays a decisive role, both for mixed-phase and cirrus clouds". The authors should state clearly than one expect biases to come either from ice behavior with respect to liquid within the mixed-phase temperature range or from ice formation at temperature below -38C, and better explain the reasons.

Define the acronyms (e.g., CALIPSO, GOCCP, COSP, CERES etc...)

P2 L18-19: The sentence could be re-phrased, "K14 found that even... among a some (or number, I believe they used 6) GCMs was not reduced".

P2 L6: I believe ice-containing clouds would be more appropriate. Why not to compare the snow water content + IWC between the REF and 2M models?

P8 L8: but may indirectly affect the cloud in the tropics, especially considering the large amount of high clouds removed

P8 L10 version night time? The authors do not explain what is a simulator at all and why using cosp here. The sentence does not tell much. The SQRT(X^2) of the bias and the correlation pattern number would help better assess the improvement of the new model version.

Fig. 3: There is no height in Fig3 Adding the contour of the difference in the original cloud cover on the bottom plot (i.e., the contour of the blue color, -5% in bottom right plot of Fig. 2) could help identifying areas of improvement. It seems like there is no change at all in middle cloud, which are lacking even in areas with no overlying high-cloud which could cause shielding effect of the lidar.

P8 L18-22: I'm not sure I understand the sentence: "The fact..." The authors state that changing microphysics does not affect CRE, that is not true (e.g., Cesana et al., 2017; their Fig. 3). The authors might get similar CREs because they tune the TOA fluxes. Also in their Fig. 4, it is clear that there are regional differences in the GCMs' CREs, i.e., over the Southern Ocean. This bias is worsened by the new GCMs, probably because of less supercooled liquid sustained in the mixed-phase clouds. The authors do

not explain why they chose these particular observation datasets. For the fluxes, I believe CERES-EBAF is the most relevant dataset for model evaluation also the longest period of time available (therefore a better climatological estimate of the present-day mean state), which is not defined either. Same thing for the cloud cover, no reason for these specific datasets and while it is mentioned that the simulator is used before (although it is not mentioned why) here no information is given whatsoever. I would recommend using only simulator-derived model outputs against GCM-oriented observation datasets, e.g., ISCCP, simulator Klein and Jakob, 1999 and dataset: Pincus et al., 2012, MODIS, simulator and dataset Pincus et al., 2012, cloudsat simulator Marchand et al., 2008 and dataset Marchand et al., 2010, CALIPSO, simulator Chepfer et al 2008; Dataset Chepfer et al., 2010. The interannual STD may be used as an uncertainty...

P8L26: I would suggest adding "In the new scheme (i.e., 2M, 4M)..." to avoid confusion.

P8L31: Again, it is not quantified at all, so hard to say. With these 2D quantities (i.e., cloud cover), it is easy to compute means, biases and correlation, so please do so and compare to CERES-EBAF.

It is striking to see how little change there is between 2M and REF in terms of cloud cover whereas the vertical cloud fractions are tremendously different. Did the authors look at the high-cloud cover as well? Can they give a hint of why such a small difference in the cloud covers? The cloud overlap may explain this.

P9L5: Again very little information is given about the observational dataset and its weaknesses/strengths.

P10 Sec. 3.6: Is the simulator used in that comparison or do the authors compare CALIPSO-GOCCP to the direct outputs of their models?

P11 Sec. 4: While I agree that the method used here to determine the origin of the

overestimation of cloud ice is good, it is not new and it has been used in the past for different topics and referred to as "tendency" (i.e., Brient et al., 2016). It is usually not possible to do so when comparing multiple models – unless a specific experiment is designed to tackle a problem and requires these such as in Brient et al. (2016) -, which is why it does not often appear in multimodel studies.

I do not understand what justify the use of so many types of clouds. The question is where does this ice come from? The answer is threefold from what I understand. Therefore, there should be three categories: Fraction of ice from heterogenous processes Fhet, from homogenous processes Fhom and from nucleation Fnuc. The total would be 100% and figures would be easier to understand.

P13L15: But how to define unrealistic pathways when no observations are available to compare to?

P14: Again, a fraction compare to the total would make more sense.

P14L20: If the simulator is used, then the same weaknesses should affect the model outputs. Also, in the mixed-phase temperature regimes, the undef-phase category can be considered as mixed-phase "likely". By using ice/total cloud frequency you are considering these undef-phase clouds as being liquid clouds, which is true in the tropics at warm temperature but unlikely at freezing temperatures. Once again, this section raises the question of whether the lidar simulator was used in Fig. 7.

One could also look at particular latitude bands to avoid the influence of these thick clouds and see whether it impacts the Phase-T relationship, e.g., in the Arctic where these clouds are less frequent.

L13: Did the authors mean sedimentation of ice at warmer temperature? i.e., the mixed-phase temperature range?

L15 I'm not sure simplifying ice category from ice crystals and snow ice to only ice can be called as an "improvement", I'd rather use the word "difference".

L18-20: No cloud bias below -35ĚŽC is shown in this paper and the biases are not well quantitatively quantified. I don't understand the expression "arguably more reasonable tuning parameters". This should be clarified.

P16 L5-6: Checking this out by changing the sedimentation overlap to random (or even minimum) overlap and running a short 1yr or even a few month simulation should be relatively easy to do and would strengthen the conclusions.

References Brient, F., Schneider, T., Tan, Z. et al., 2016 : Shallowness of tropical low clouds as a predictor of climate models' response to warmin, Clim Dyn 47: 433. https://doi.org/10.1007/s00382-015-2846-0

Cesana G., K. Suselj and F. Brient, 2017: On the dependence of cloud feedbacks on physical parameterizations in WRF aquaplanet simulations, Geophys. Res. Lett., 44, 10,762–10,771, https://doi.org/10.1002/2017GL074820

Chepfer H., S. Bony, D. Winker, G. Cesana, JL. Dufresne, P. Minnis, C. J. Stubenrauch, S. Zeng, (2010), The GCM Oriented Calipso Cloud Product (CALIPSO-GOCCP),J. Geophys. Res., doi: 10.1029/2009JD012251

Chepfer, H., S. Bony, D. M. Winker, M. Chiriaco, J.-L. Dufresne, and G. Seze, 2008: Use of CALIPSO lidar observations to evaluate the cloudiness simulated by a climate model, Geophys. Res. Lett., 35, L15704, doi:10.1029/2008GL034207.

Klein, S.A. and C. Jakob, 1999: Validation and Sensitivities of Frontal Clouds Simulated by the ECMWF Model. Mon. Wea. Rev., 127, 2514–2531,https://doi.org/10.1175/1520-0493(1999)127<2514:VASOFC>2.0.CO;2

Pincus, R., S. Platnick, S.A. Ackerman, R.S. Hemler, and R.J. Patrick Hofmann, 2012: Reconciling Simulated and Observed Views of Clouds: MODIS, ISCCP, and the Limits of Instrument Simulators. J. Climate, 25, 4699–4720, https://doi.org/10.1175/JCLI-D-11-00267.1

Marchand, R. T., J. Haynes, G.G. Mace, T. Ackerman, and G. Stephens, 2010: A

comparison of simulated cloud radar output from the multiscalemodeling framework global climate model with CloudSat cloud radar observations, J. Geophys. Res., 114, D00A20, doi:10.1029/2008JD009790.

Marchand, R. T., G. G. Mace, and T. P. Ackerman, 2008: Hydrometeor detection using CloudSat-an earth orbiting 94 GHz cloud radar. J. Atmos. Oceanic. Technol., 25, 519-533, doi: 10.1175/2007JTECHA1006.1.

---

## Referee Comment (RC2) · Anonymous Referee #2 · 6 Sep 2018

Dear Editor, dear Authors,

I have reviewed "Elucidating ice formation pathways in the aerosol–climate model ECHAM6-HAM2" by Dietlicher et al. The manuscript presents a method to quantify the origin of ice and liquid condensate in clouds. The authors use this method to classify ice origin by homogeneous or heterogeneous freezing and conclude that the high ice bias in mixed-phase clouds in ECHAM-HAM is mostly due to sedimentation of ice that formed by homogeneous freezing.

The manuscript is well written, and the result is both interesting and important. I recommend acceptance of the manuscript.

There are a few questions that the manuscript did not address and that the authors

might consider clarifying in the final version.

- I found it a bit confusing that the paper tries to do two very different things at once: (1) present validation of a new ice microphysics parameterization (that has already been described in a GMD article) against observations and (2) introduce new tracers to classify the origin of cloud ice, a technique that is applicable to new and old microphysics alike. Scientifically, the second part of the paper is far more interesting, and I feel the first part might have found a better home in the GMD paper. Perhaps there is a way to tie the two parts together a bit more in Sec. 5.2, by describing whether there are significant differences between the new and old microphysics, and in particular whether the new microphysics leads to an improvement. (I realize Fig. 7 does this for the state, but I don't see analogous discussion for the pathways.)

- I agree with the sentiment of the introductory paragraph of Sec. 4 (although I would make an exception for observations that permit inference of process rates or the relative importance of various processes). Of course, this paragraph comes right after a long section that does the exact thing the authors criticize. Perhaps this is an argument in favor of shortening Sec. 3 or moving parts of it to an appendix?

- The previous point notwithstanding, in Sec. 3 (Tab. 3 in particular), I was surprised that the authors provide an uncertainty range for radiative flux observations but not for the ice water path. IWP seems like the more directly relevant variable to evaluate the ice microphysics scheme. It would be nice to see whether passive microwave, MODIS, etc. IWP estimates are as far away from the model as CloudSat/Calipso. Also, why not add the TIWP in the REF model to Tab. 3 under the assumption that the sedimentation occurs within the time step? (And likewise for CIWP in the new configuration?)

- In the discussion of deposition acting as a sink for cloud cover via the Sundqvist cloud cover scheme (Sec. 2.2), I would have welcomed a sentence or two on whether condensation analogously acts as a sink for cloud cover or how this is avoided. Also, the sentence "However, this coupling also makes the sedimentation sink of cloud ice a sink for cloud fraction" made me wonder: isn't that realistic, desirable behavior?

- Sec. 3.2, better agreement with GOCCP cloud cover: was this part of the tuning strategy, or did it emerge?

- Sec. 4.3, last sentence: would "cirrus-origin cloud" be less confusing terminology than "cirrus"?

- Sec. 5.1, Fig. 10: The frequencies here are defined by volume. If they were defined by mass, which I assume would be equally valid but give greater weight to warmer clouds, would the conclusions be very different?

- Sec. 5.2, l. 19–21: This seems out of place here; maybe a better place would be in Sec. 3.6?

-

A few minor typos etc:

- p. 1, l. 15: "radiative forcing" → "radiative effect", since the clouds are part of the climate system?

- p. 2, l. 21: I kept wondering for the rest of the manuscript why the homogeneous freezing threshold is $-35$ rather than $-38°$ C.

- p. 3, l. 3: "eluded" → "alluded".

- p. 3, l. 24: Can you comment on how applicable this is to other models?

- p. 4, l. 32: "lead" → "led".

- p. 7, l. 27: "does no longer require" → "no longer requires".

- p. 8, l. 23: "an" → "and".

- p. 13, l. 7: "areaf" → "are".

- p. 13, l. 15: "thereof" → "therefrom".

- p. 15, l. 18: "does no longer have" → "no longer has".

- Tab. 2 uses "QSW" and "HCl", which I assume are meant to be "LIM_ICE" and "HET_CIR".

- Fig. 2: Only color scale for differences is included in the plot.

- Koop et al. (2000): citation is missing a DOI.

- Platnick et al. (2017): citation data appears incomplete.

---

## Author Comment (AC1) · 1 Nov 2018

**Reply to anonymous Referee #1**

**Remo Dietlicher**

**November 1, 2018**

Thank you for carefully reading our manuscript. We are happy for your expertise regarding satellite data. In the following we answer the individual points you raise:

Main comments: In the model evaluation section, the authors do not describe the dataset at all and do not explain why they chose these specific datasets to evaluate their model. For example, three datasets are used for the fluxes and no reason whatsoever is given to justify this choice. The authors must define/introduce the datasets, even briefly, and explain why they use these. Also, the observed interannual STD can be used as an uncertainty estimates when nothing else is available. In addition, their model evaluation is more of a qualitative comparison than a quantitative one although it is possible to quantify the bias more precisely (see specific comments for further details). It also looks like (it is not specified in the manuscript) they didnt make a consistent comparison between the CALIPSO-GOCCP cloud phase dataset and their model outputs, i.e., they didnt use the simulator for the cloud phase diagnostics, which make the results difficult to interpret. In the second part of the manuscript, I feel like reducing the number of categories in the final part of the study would help to better determine the origin of the ice bias. Id like the author to either do that or better explain why they chose these categories and what is the added value of making this choice. Finally, the authors could easily check at least one of the mechanism that supposedly lead to the overestimation of the ice cloud occurrences (the overlap assumption).
(Introduction): The goal of the paper is a little bit confused and not clearly stated. Also, it is not clear to me how 'As has been eluded to above, the formation history of a cloud plays a decisive role, both for mixed-phase and cirrus clouds'. The authors should state clearly than one expect biases to come either from ice behavior with respect to liquid within the mixed-phase temperature range or from ice formation at temperature below -38C, and better explain the reasons.
We have restructured the introduction to better motivate the study. We now highlight that the cloud phase partitioning is governed by the ice phase parametrizations and that we want to figure out which process dominates in our model.

Define the acronyms (e.g., CALIPSO, GOCCP, COSP, CERES etc...)
Done.

P2 L18-19: The sentence could be re-phrased, K14 found that even... among a some (or number, I believe they used 6) GCMs was not reduced.
Done. Actually when checking the exact models, it was interesting to see so many models from the CAM family.

P2 L6: I believe ice-containing clouds would be more appropriate. Why not to compare the snow water content + IWC between the REF and 2M models?

We diagnose the snow mass flux in REF, assuming that all snow will reach the ground within one model timestep. Therefore we can quantify the column-integrated amount of snow but there is no way to assign snow contents to individual levels.

P8 L8: but may indirectly affect the cloud in the tropics, especially considering the large amount of high clouds removed

This is true, we adjusted the text.

P8 L10 version night time? The authors do not explain what is a simulator at all and why using cosp here. The sentence does not tell much. The SQRT(X2) of the bias and the correlation pattern number would help better assess the improvement of the new model version.

We use day and night. We extended this paragraph to motivate the usage of COSP better. We now also compute the Pearson correlation coefficient and RMSE between the models and CALIPSO to allow for a more quantitative discussion.

Fig. 3: There is no height in Fig3 Adding the contour of the difference in the original cloud cover on the bottom plot (i.e., the contour of the blue color, $-5\,\%$ in bottom right plot of Fig. 2) could help identifying areas of improvement. It seems like there is no change at all in middle cloud, which are lacking even in areas with no overlying high-cloud which could cause shielding effect of the lidar.

The height axis must have disappeared by mistake, its in there again. Thank you for noticing this. Adding the $-5\,\%$ contour line to highlight areas where the new cloud cover parameterization acts is a good idea. The new scheme has been designed to make the transition from mixed-phase to cirrus clouds continuous and consistent with the parameterization of the formation of cirrus clouds. Improving the cirrus cloud structure is a nice side-product. Improving the mid-level cloud structure has not been the focus of this study.

P8 L18-22: Im not sure I understand the sentence: The fact... The authors state that changing microphysics does not affect CRE, that is not true (e.g., Cesana et al., 2017; their Fig. 3). The authors might get similar CREs because they tune the TOA fluxes. Also in their Fig. 4, it is clear that there are regional differences in the GCMs CREs, i.e., over the Southern Ocean. This bias is worsened by the new GCMs, probably because of less supercooled liquid sustained in the mixed-phase clouds. The authors do not explain why they chose these particular observation datasets. For the fluxes, I believe CERES-EBAF is the most relevant dataset for model evaluation also the longest period of time available (therefore a better climatological estimate of the present-day mean state), which is not defined either. Same thing for the cloud cover, no reason for these specific datasets and while it is mentioned that the simulator is used before (although it is not mentioned why) here no information is given whatsoever. I would recommend using only simulator-derived model outputs against GCM-oriented observation datasets, e.g., ISCCP, simulator Klein and Jakob, 1999 and dataset: Pincus et al., 2012, MODIS, simulator and dataset Pincus et al., 2012, cloudsat simulator Marchand et al., 2008 and dataset Marchand et al., 2010, CALIPSO, simulator Chepfer et al 2008; Dataset Chepfer et al., 2010. The interannual STD may be used as an uncertainty...

We have rewritten this paragraph to make clear that CRE is a consequence of tuning

TOA fluxes. The fact that it is more negative than what observations suggest hints at a structural problem in the model that is not specific to the microphysics scheme.
Regarding the TOA fluxes in Fig. 4, we agree that the original Figure was confusing. We now only use the CERES-EBAF dataset as suggested and plot the interannual STD as a measure of uncertainty. Furthermore, we compute the correlation and RMSE of the full 2D fields. We agree that these statistics provide interesting and important information for a more quantitative assessment of these fundamental model variables. This analysis revealed that the statement about the new model correlating better with the observations was false, even though the zonal mean suggested that.

P8L26: I would suggest adding In the new scheme (i.e., 2M, 4M)... to avoid confusion.
We replaced this sentence with something more precise.

P8L31: Again, it is not quantified at all, so hard to say. With these 2D quantities (i.e., cloud cover), it is easy to compute means, biases and correlation, so please do so and compare to CERES-EBAF.
This has been addressed in a previous comment.

It is striking to see how little change there is between 2M and REF in terms of cloud cover whereas the vertical cloud fractions are tremendously different. Did the authors look at the high-cloud cover as well? Can they give a hint of why such a small difference in the cloud covers? The cloud overlap may explain this.
The new and reference models differ most in high-level clouds and are fairly similar for mid- and low-level clouds. Since both models tend to underestimate the cloud fraction, the overestimation of the high-level cloud fraction in the reference model improves the total cloud cover in areas where the cloud fraction would be small otherwise. We agree that the reason for the smaller difference among the models in terms of total cloud cover as compared to the vertical structure is due to vertical overlap. This is now mentioned at the end of Section 3.2.

P9L5: Again very little information is given about the observational dataset and its weaknesses/strengths.
We extended this paragraph to motivate the use of the Li et al, 2012 dataset better.

P10 Sec. 3.6: Is the simulator used in that comparison or do the authors compare CALIPSO-GOCCP to the direct outputs of their models?
We do not use a simulator for Fig. 7 but added a new Fig. 8 including output from the COSP simulator. More details follow below.

P11 Sec. 4: While I agree that the method used here to determine the origin of the overestimation of cloud ice is good, it is not new and it has been used in the past for different topics and referred to as tendency (i.e., Brient et al., 2016). It is usually not possible to do so when comparing multiple models unless a specific experiment is designed to tackle a problem and requires these such as in Brient et al. (2016) -, which is why it does not often appear in multimodel studies.
We changed the text to highlight the reason why this diagnostic is very helpful to answer the specific question at hand 'where does ice come from?' and better differentiate this method from analyzing model tendencies. In a nutshell, tendencies are a snapshot of

the strength of processes but provide no history, which source-tagged tracers do. However, implementing our method requires additional prognostic tracers, a substantial effort, which makes it even more unlikely to be used in model inter-comparisons.

I do not understand what justify the use of so many types of clouds. The question is where does this ice come from? The answer is threefold from what I understand. Therefore, there should be three categories: Fraction of ice from heterogenous processes Fhet, from homogenous processes Fhom and from nucleation Fnuc. The total would be 100 % and figures would be easier to understand.

The goal of the cloud types defined in this study is to differentiate clouds with fundamentally different properties. A priori, we did not know what to expect but wanted to include all information that is available in the model. For example, we were interested to know whether mixed-phase clouds are so rare because (1) mixed-phase freezing occurs infrequently or (2) whether the subsequent ice growth is slow. Distinguishing ice and liquid dominated mixed-phase clouds allows quantifying both aspects: (1) mixed-phase freezing is only important in a small fraction of clouds and (2) the effect it has on the cloud phase partitioning is even smaller because a majority of mixed-phase clouds do not (or slowly) glaciate. Similarly, including the vertical cloud structure allowed to diagnose sedimentation in vertically adjacent cloudy layers as the relevant pathway for the transport of ice into the mixed-phase regime. A priori, it could also have been the case that e.g. sublimation in clear-sky levels is underestimated such that ice can pass too many subsaturated layers. Finally, we differentiate warm and cold liquid clouds to quantify the cloud amount that could be affected by freezing.
The sum of all cloud types is exactly the total cloud cover.

P13L15: But how to define unrealistic pathways when no observations are available to compare to?
We have rewritten this paragraph to be more precise about the use of our method and refrain from the word 'unrealistic' which we agree cannot be assessed by observations.

P14: Again, a fraction compare to the total would make more sense.
We assume this comment is regarding Fig 12 which is discussed on page 14. Having both plots at hand, we believe there is no benefit of normalizing by total cloud cover. Qualitative statements about the relative contribution to the total cloudiness can still easily be made.

P14L20: If the simulator is used, then the same weaknesses should affect the model outputs. Also, in the mixed-phase temperature regimes, the undef-phase category can be considered as mixed-phase likely. By using ice/total cloud frequency you are considering these undef-phase clouds as being liquid clouds, which is true in the tropics at warm temperature but unlikely at freezing temperatures. Once again, this section raises the question of whether the lidar simulator was used in Fig. 7.
Since we are not using a simulator for Fig. 7, we added a new Fig. 8 which compares the phase ratios versus temperature lines from the satellite, the model and the model + simulator. This was actually an interesting and worthwhile exercise. The overestimation at $T > 15\,°C$ is much reduced when using a simulator, implying that attenuation is very important in this temperature regime. It makes sense, since the thick cloud type is also optically thick, the lidar signal will always be attenuated in the lower part of these clouds.

Unfortunately, this implies that the satellite cannot be used to constrain the cloud phase partitioning in this temperature regime.

Figure 7 was inspired by Cesana et al. 2015 where they conduct a comprehensive model inter-comparison of phase ratio vs. temperature histograms (their Fig. 10). For the satellite panel we reproduced their Fig. 10 by using daily night-time data from here: ftp://ftp.climserv.ipsl.polytechnique.fr/cfmip/GOCCP_v3/3D_CloudFraction/grid_2x2xL40/2008/night/daily for the years 2008-2014. These files contain a variable called *cltemp_phase* which is computed as ice/(ice+liquid), i.e. neglects pixels where no phase can be assigned (undef). This corresponds exactly to the computation within the COSP simulator.

One could also look at particular latitude bands to avoid the influence of these thick clouds and see whether it impacts the Phase-T relationship, e.g., in the Arctic where these clouds are less frequent.

Thick and cirrus clouds are the dominant cloud type almost everywhere on the globe (see Fig. 11).

L13: Did the authors mean sedimentation of ice at warmer temperature? i.e., the mixed-phase temperature range?

We meant to say from colder (below $-35\,°\mathrm{C}$) to warmer temperatures (i.e. the mixed-phase regime). This is now stated more clearly.

L15 Im not sure simplifying ice category from ice crystals and snow ice to only ice can be called as an improvement, Id rather use the word 'difference'.

It is an improvement in terms of the physical realism of the scheme at the cost of increased computational demand because sedimentation of ice crystals needs to be resolved. So improvement might indeed be a little bit too general and we replaced it with 'difference'.

L18-20: No cloud bias below $-35\,°\mathrm{C}$ is shown in this paper and the biases are not well quantitatively quantified. I dont understand the expression 'arguably more reasonable tuning parameters'. This should be clarified.

We show that the reference model overestimates the cloud fraction at temperatures below $-35\,°\mathrm{C}$ in Fig. 3. In the updated version of the manuscript we now compute the correlation and RMSE and are confident that the new cloud cover parametrization is not just a conceptual improvement but also leads to a more realistic cloud fraction for these high clouds. We removed the sentence about tuning parameters. Even though we are more happy with scaling process rates by a factor of 5 rather than 1000, they are fundamentally unconstrained so there is not really a metric to assess good and bad.

P16 L5-6: Checking this out by changing the sedimentation overlap to random (or even minimum) overlap and running a short 1yr or even a few month simulation should be relatively easy to do and would strengthen the conclusions.

Given all the changes above, the part that is referenced here has been removed.

---

## Author Comment (AC2) · 1 Nov 2018

**Reply to anonymous Referee #2**

**Remo Dietlicher**

**November 1, 2018**

Thank your for carefully reading our manuscript and the positive review. Below we elaborate on the points that you mention and outline how we addressed them.

I found it a bit confusing that the paper tries to do two very different things at once: (1) present validation of a new ice microphysics parameterization (that has already been described in a GMD article) against observations and (2) introduce new tracers to classify the origin of cloud ice, a technique that is applicable to new and old microphysics alike. Scientifically, the second part of the paper is far more interesting, and I feel the first part might have found a better home in the GMD paper. Perhaps there is a way to tie the two parts together a bit more in Sec. 5.2, by describing whether there are significant differences between the new and old microphysics, and in particular whether the new microphysics leads to an improvement. (I realize Fig. 7 does this for the state, but I dont see analogous discussion for the pathways.)

We completely agree that the validation part of this paper would have fit also together with the technical evaluation in the GMD paper. However, we chose this composition to segregate the idealized single column simulations which highlight the technical aspects of the new scheme from the global evaluation presented here.
Technically, one big improvement that the new microphysics scheme brings is a more readable and manageable code-base. This allowed to easily implement the formation pathway diagnostics. Porting this to the old microphysics code would have been a considerable effort which is why we cannot compare the pathway analysis between the models. The touching point is Fig. 7 where we see a similar phase ratio and thus assume that probably the same mechanisms are in place. Especially since we could trace the high frequency of ice clouds at high temperatures back to the vertical structure of clouds (thick category) which is a result of parametrizations that are similar or even identical in the two models. We briefly addressed this problem now in the introduction to Section 5.

I agree with the sentiment of the introductory paragraph of Sec. 4 (although I would make an exception for observations that permit inference of process rates or the relative importance of various processes). Of course, this paragraph comes right after a long section that does the exact thing the authors criticize. Perhaps this is an argument in favor of shortening Sec. 3 or moving parts of it to an appendix?

We don't see a viable way to evaluate GCM output other than comparing to climatologies derived from (satellite-)observations. Comprehensive case-studies which allow to infer microphysical process rates usually only target specific clouds and meteorological conditions which cannot easily be generalized to be used in a GCM. We therefore don't want do abandon spatio-temporally averaged model output but rather highlight the fact that in this kind of output a lot of valuable information is lost. In light of your comment, we

have rewritten this part of the introductory paragraph of Section 4 to be more precise.

The previous point notwithstanding, in Sec. 3 (Tab. 3 in particular), I was surprised that the authors provide an uncertainty range for radiative flux observations but not for the ice water path. IWP seems like the more directly relevant variable to evaluate the ice microphysics scheme. It would be nice to see whether passive microwave, MODIS, etc. IWP estimates are as far away from the model as CloudSat/Calipso. Also, why not add the TIWP in the REF model to Tab. 3 under the assumption that the sedimentation occurs within the time step? (And likewise for CIWP in the new configuration?)

You are right, it makes a lot of sense to include the uncertainty range for T/CIWP in Table 3. It has been added. We agree that IWP/C is the most relevant variable to evaluate the new (ice) microphysics scheme. Nevertheless, a direct comparison remains difficult due to uncertainties in the retrievals and the heterogeneous representation of ice in models. In our reference model ice is split up into in-cloud ice and stratiform and convective snow. The new model can uniformly describe stratiform precipitation but the uncertainty from convective ice still remains.

We do not think that we can use a 'diagnostic trick' to homogenize model output. The reference model diagnoses the snow mass flux as $P_{snow} = \int_0^{p_s} Sources - Sinks\, dp$ for the surface pressure $p_s$. The mass mixing ratio of snow therefore relies on the sedimentation velocity of snow which is rather uncertain since there is no prognostic information on the snow particle size. Similarly, computing CIWP for the new model would require a threshold size or fall velocity above which ice crystals are considered to be snow. This goes directly against a main benefit of the P3 scheme which is eliminating such threshold sizes.

In the discussion of deposition acting as a sink for cloud cover via the Sundqvist cloud cover scheme (Sec. 2.2), I would have welcomed a sentence or two on whether condensation analogously acts as a sink for cloud cover or how this is avoided. Also, the sentence 'However, this coupling also makes the sedimentation sink of cloud ice a sink for cloud fraction' made me wonder: isn't that realistic, desirable behavior?

For cloud water we do not have this problem as condensation/evaporation is given by Eq. (2) which is a form of saturation adjustment and does not allow supersaturation w.r.t. liquid water by definition. There is only a problem for cloud ice which either forms from a liquid cloud or nucleates directly from the vapor phase. Both pathways require substantial supersaturation w.r.t. ice. Therefore we need to specify what happens once the initial ice crystals have formed.

Regarding your second point we agree in principle. Our concern with this is mostly the increase of in-cloud ice crystal number concentrations and the resulting feedback loop involving the coupling of aggregation, sedimentation and cloud cover. This is discussed in Section 3.4 paragraph 3 where we argue that it explains the lower cloud cover found in the 2M as compared to LIM_ICE simulation.

Sec. 3.2, better agreement with GOCCP cloud cover: was this part of the tuning strategy, or did it emerge?

The main goal of the new cloud cover parametrization was to consistently extend the notion of the subgrid cloud fraction to the cirrus regime. In the reference model there is a mismatch between how the cloud fraction is diagnosed ($b = 1$ at ice saturation) and the cirrus cloud formation processes (efficient nucleation only at $RH =\sim 140\,\%$). Tying the

cloud cover and cirrus nucleation parametrizations together there is no freedom to tune the parametrization. We now highlight this in Section 3.2.

Sec. 4.3, last sentence: would 'cirrus-origin cloud' be less confusing terminology than cirrus?
This is a philosophical question that came up in the process of this project as well. In my eyes, a cirrus cloud does not lose its 'cirrus'-ness when it crosses a certain temperature threshold. I also like to be very cautious when using real-life intuition on model output. These readily sedimenting cirrus clouds seem to be much more prominent in the model-world than in real-life. So if anything, we could call it 'model cirrus' but then again I guess the 'model' part is implied.

Sec. 5.1, Fig. 10: The frequencies here are defined by volume. If they were defined by mass, which I assume would be equally valid but give greater weight to warmer clouds, would the conclusions be very different?
They are actually defined by air mass, not volume. We refrained from calling it cloud mass as this could be confused with the mass of cloud condensate which would drastically alter the relative contributions. We do not believe that there is a substantial change if we use air volume or mass. If we were to use volume the relative contribution of cirrus would probably be somewhat higher.

Sec. 5.2, l. 19-21: This seems out of place here; maybe a better place would be in Sec. 3.6?
You are right in that this would fit nicely in Section 3.6 when Fig. 7 is discussed. However, we need to introduce the formation pathways (Section 4) before we can discuss the different origins of cloud ice. We extended the introductory paragraph to Section 5 to better tie the two parts together.

p. 1, l. 15: 'radiative forcing' $\rightarrow$ 'radiative effect', since the clouds are part of the climate system?
You are right!

p. 2, l. 21: I kept wondering for the rest of the manuscript why the homogeneous freezing threshold is $-35\,°C$ rather than $-38\,°C$.
In theory the threshold is close to $-38\,°C$. However, our model traditionally used $-35\,°C$ (based on Lohmann and Roeckner, 1996) as a freezing threshold which is why we use this when talking about the model. However, this number does no longer appear in any of the parametrizations of the new model since homogeneous freezing and cloud cover parametrizations have been replaced from the reference model (where a threshold value of $-35\,°C$ is used).

p. 3, l. 24: Can you comment on how applicable this is to other models?
Since it requires solving additional prognostic tracers according to Eqs. (5), (6) and (A1), it is probably unrealistic to implement in a large model intercomparison. However, these tracers are easily implemented if the cloud tendencies are accessible in the code ($S$-terms in the equations). This is why it is hard to do for our reference model where the computation of tendencies is often intertwined with local variable updates.

138   Fig. 2: Only color scale for differences is included in the plot.

139   This is not the case in my PDF-viewer.. Something to double-check when type-setting.

140

141   Thank you for finding various typos, they have all been corrected.

---

## Referee Report (RR1)

**Decision**

In this new version of the manuscript, the authors have addressed part of my concerns by documenting more carefully the observational datasets and by clarifying the goal of their study. However, they still compare directly the model outputs to satellite observations in section 3.6 - making it an apples-to-oranges comparison - and draw wrong conclusions based on this. Therefore, I would recommend revisiting this section and updating the conclusions accordingly – along with a few minor things – before publication.

**Main comments:**

There is a confusion in the paper between optically thick and geometrically thick although I reckon that 500mb thick clouds may also correspond to optically thick clouds. This should be clarified in the manuscript.

Additionally, my main concern is about Section 3.6. While I acknowledge that the authors produce a lot of efforts to improve this section, it is still confusing for the reader and needs further work. In addition, I disagree with some conclusions made by the authors that are not supported by evidences.

At the beginning of the section, the authors state that the comparison is done using the simulator. However, the first figure doesn't show any simulator results and most of the conclusions are (wrongly) made based on this particular figure. It's even more confusing as the authors state "here we use COSP simulator…" right after mentioning Fig. 7. The reader is left thinking that he can compare all plots in Fig. 7 in a consistent way.

Yet, the differences shown here are due to an apples-to-oranges comparison: "mass phase ratio vs. frequency phase ratio". The satellite observations should be compared with the model output+ simulator only. The authors have the model+simulator outputs so they should add a row to Fig. 7, reproduce the top row using the model+simulator outputs and move the observations to that bottom row (and delete fig. 8). In addition, as mentioned in the previous round of review, the authors should treat the undefined-phase clouds as being mixed-phase clouds. Their lidar return as well as their mixing ratio are dominated by the ice phase, which is why they should be accounted for in the ice type of clouds (e.g., Cesana et al., 2016).

From this section and section 5, the authors conclude that "*the formation pathways revealed that most of the simulated ice clouds are in the blind spot of the lidar in the lower part of optically thick clouds*". This statement is not supported by evidences. If the authors want to find out about this, they should i) filter out these clouds of the statistics and ii) select them only to reproduce fig. 7 in both the model outputs and model+simulator outputs. Only then, the authors would be able to conclude what is the effect of these clouds on the MPR/FPR vs T relationship. However, these conclusions would be valid only for this particular model.

At the moment, nothing allows one to conclude that these thick clouds (less than 30% of all clouds) are responsible for the differences between model outputs and model+sim outputs and even less that the bottom part of these clouds is.

To further show that these clouds probably do not affect much the FPR-T relationship I provide below a brief analysis using the observation. CALIPSO-GOCCP now offer new information about the opacity of the column. It is possible to discriminate lidar shots that reach the ground from those completely attenuated (Guzman et al., 2017), the latter being less frequent than the former. After computing the ratio of thin cloud to thin+opaque clouds, I reproduced Fig. 8 for all clouds (black-dash line) thin-cloud dominated gridboxes (ratio>0.5, magenta dash-line) and opaque-cloud dominated gridboxes (ratio<0.5, green-dash line). Undefined-phase clouds are threated as ice clouds (as mentioned earlier). This highlights the very small impact of *optically thick* clouds on the FPR-T relationship.

[Figure]

*Figure 1: Reproduced Fig. 8 with CALIPSO-GOCCP FPR for all (black-dash), thin-dominated (magenta-dash) and opaque-dominated (green-dash) clouds. Note that the solid black line is different from the black-dash line because the undefined-phase clouds are treated as ice clouds: FPR is the ratio between ice+undefined-phase clouds to ice+undefined-phase + liquid clouds. This plot was created using 2x2˚ monthly CALIPSO-GOCCP observations version 3.1.2, nighttime only, for the period 2007-2016 (Map_OPAQ and 3D-CloudFraction_Temp files).*

**Minor comments:**
P1 L17: The authors do not demonstrate this in their study, they just show that the mixing phase ratio is substantially different from the frequency phase ratio in their model, and that roughly 30% of the clouds are "geometrically" thick clouds in their model. This doesn't mean that the differences MPR/FPR are due to the bottom of the geometrically thick clouds (called optically in the abstract).

P2 L3: Community ➔ Cloud

P9 L13: make "use" of…

P9 L20: from 2006-2012? In the caption of Fig. 3 it is written 2008-2014.

P9 L24-28: anomalies ➔ bias
I guess you refer to the change from blue to red while stating "distinct jump". It might be good to state this in the text.

P9 L28 "is differ less"

P9 L29: This actually shown in Cesana and Waliser, 2016. If the overlap were in better agreement with the observations, the cloud amount overestimation would be substantially larger, such as that found in Fig. 3.

Fig 4 & 5: the 2M lines are difficult to spot because all other lines are on top of it. It might be worth it to replot this with 2M on top.
When I suggested to add the +/- 1STD I meant annual STD, so that the frequency of annual oscillation could be taken into account when compared to the model. I believe the authors show here a monthly interannual STD rather than a yearly interannual STD. I'd recommend that the authors change the monthly interannual STD to yearly interannual STD in these figures. I did it myself for the LW/SW CRE and the results is far smaller than what shown here.

P13 L13: it is not based "on a threshold value for the backscattered polarization ratio to separate…" but on the cross-polarized attenuated backscatter and the total attenuated backscatter and a discrimination line. Note that this process is reproduced in the simulator
P13 L14-L15: The lidar spatial resolution is actually around 70m while the distance between lidar shots is ~333m.

P13 L33: They are more frequent in the 2M simulation than the REF and it should be stated. While not comparable, it is also much more frequent in the observations.

P14 L1: I don't understand this statement.

P19 L17-18: This is wrong, it is due to comparing two different quantities, which do not take into account the instrument peculiarities. One should only conclude based on comparison between obs and model+sim outputs.

P19 L18-19: As explained before, this is not what's shown in the paper.

P19 L19-20: I completely disagree with this statement, which is not supported by any evidence. Even if it was shown that the PR-T definition is substantially different in ECHAM6 for thick and thin clouds (which is not performed so far), it would only suggest that the phase simulated with the lidar simulator cannot be used to assess the PR-T relationship of all clouds in this particular model.

---

## Author Response (AR2)

**Reply to anonymous Referee #1**

Remo Dietlicher

March 15, 2019

4 Thank you for going through our manuscript again in detail. We much appreciate your
5 concerns which enable us to further improve the clarity of this manuscript. Your main
6 concern about the comparison of the simulated and observed phase ratios has been re7 solved. We now only compare the average phase ratio per temperature from the lidar
8 simulator to the satellite retrieval. The PR-T plot of the CALIPSO-GOCCP dataset
9 next to the direct model output has been removed as it was misleading.

**10 1 Major comments**

There is a confusion in the paper between optically thick and geometrically thick although
I reckon that 500mb thick clouds may also correspond to optically thick clouds. This
should be clarified in the manuscript.

We changed the name of this category from *thick* to *deep* which is more accurately captures the geometrical definition of this cloud type.

16

1

2

3

Additionally, my main concern is about Section 3.6. While I acknowledge that the authors
produce a lot of efforts to improve this section, it is still confusing for the reader and needs
further work. In addition, I disagree with some conclusions made by the authors that are

20 not supported by evidences.

Section 3.6 first and foremost presents the modeled phase ratio histogram. In the following 21 sections we introduce and apply a formation pathway analysis to dissect this histogram 22 to provide a clear picture of how and where ice forms. For the modelling community, 23 this is of great importance since there is a wide spread in the modeled PR-T relation-24 ship, the controls for which are hard to find. With that in mind, the use of a simulator 25 is not feasible as it will blur the view on the modeled cloud field. For our purpose, we 26 therefore rather remove the disputed third plot in Fig. 7 and only compare the model 27 to observations in the T-space in Fig. 8. Given the strong bi-modality of the modeled 28 PR-T distribution, there is almost no information lost. We changed the units and the 29 style of Fig. 13 to highlight the relationship between Fig. 13 (average cloud type frac-30 tion per temperature) and Fig. 8 (average phase ratio per temperature). The disputed 31 conclusion about the origin of the simulator+model to model difference has been removed. 32 33

At the beginning of the section, the authors state that the comparison is done using the simulator. However, the first figure doesn't show any simulator results and most of the conclusions are (wrongly) made based on this particular figure. Its even more confusing as the authors state here we use COSP simulator... right after mentioning Fig. 7. The reader is left thinking that he can compare all plots in Fig. 7 in a consistent way. Yet, the differences shown here are due to an apples-to-oranges comparison: mass phase ratio vs. frequency phase ratio. The satellite observations should be compared with the model output+ simulator only. The authors have the model+simulator outputs so they should add a row to Fig. 7, reproduce the top row using the model+simulator outputs and move the observations to that bottom row (and delete fig. 8).

44 This part has been refactored to better with the main story of the manuscript (motivate

45 Fig. 13). The confusing use of COSP has been removed as discussed above.

46

47 In addition, as mentioned in the previous round of review, the authors should treat the 48 undefined-phase clouds as being mixed-phase clouds. Their lidar return as well as their 49 mixing ratio are dominated by the ice phase, which is why they should be accounted for 50 in the ice type of clouds (e.g., Cesana et al., 2016).

This has been done for both the COSP simulator (where the effect is very small) and the satellite product.

53

From this section and section 5, the authors conclude that the formation pathways re-54 vealed that most of the simulated ice clouds are in the blind spot of the lidar in the lower 55 part of optically thick clouds. This statement is not supported by evidences. If the au-56 thors want to find out about this, they should i) filter out these clouds of the statistics and 57 ii) select them only to reproduce fig. 7 in both the model outputs and model+simulator 58 outputs. Only then, the authors would be able to conclude what is the effect of these 59 clouds on the MPR/FPR vs T relationship. However, these conclusions would be valid 60 only for this particular model. At the moment, nothing allows one to conclude that these 61 thick clouds (less than 30% of all clouds) are responsible for the differences between model 62 outputs and model+sim outputs and even less that the bottom part of these clouds is. 63

64 This statement has been removed

65

To further show that these clouds probably do not affect much the FPR-T relationship I 66 provide below a brief analysis using the observation. CALIPSO-GOCCP now offer new 67 information about the opacity of the column. It is possible to discriminate lidar shots that 68 reach the ground from those completely attenuated (Guzman et al., 2017), the latter being 69 less frequent than the former. After computing the ratio of thin cloud to thin+opaque 70 clouds, I reproduced Fig. 8 for all clouds (black-dash line) thin-cloud dominated gridboxes 71 (ratio>0.5, magenta dash-line) and opaque-cloud dominated gridboxes (ratio<0.5, green-72 dash line). Undefined-phase clouds are threated as ice clouds (as mentioned earlier). This 73 highlights the very small impact of optically thick clouds on the FPR-T relationship. 74 While it is very nice to see that there are observational constraints on this issue, there is 75 no reason to believe that the model must behave exactly the same way as the retrieval. 76 In fact, this study as well as Cesana and Waliser (2016), both suggest that the vertical 77

restent of clouds is overestimated in some models, most likely enhancing the effect these

r9 (more frequent) opaque clouds have on the phase ratio as compared to what your satellite

80 analysis suggests.

**81 2 Minor comments**

P1 L17: The authors do not demonstrate this in their study, they just show that the mixing phase ratio is substantially different from the frequency phase ratio in their model, and that roughly 30% of the clouds are geometrically thick clouds in their model. This

doesn't mean that the differences MPR/FPR are due to the bottom of the geometrically 85 thick clouds (called optically in the abstract). 86 This statement has been removed as it does not contribute to the main message of the 87 manuscript. 88 89 P2 L3: Community  $\rightarrow$  Cloud 90 Turns out its a model intercomparison project for 'coupled' models. 91 92 P9 L13: make use of... 93 Thanks, this is corrected. 94 95 P9 L20: from 2006-2012? In the caption of Fig. 3 it is written 2008-2014. 96 This is wrong in the text, it is data for 2008-2014 and has been corrected. 97 98 P9 L24-28: anomalies  $\rightarrow$  bias 99 I guess you refer to the change from blue to red while stating distinct jump. It might be 100 good to state this in the text. 101 This sentence has been rephrased to be more concise. 102 103 P9 L28 'is differ less' 104 The 'is' has been removed. 105 106 P9 L29: This actually shown in Cesana and Waliser, 2016. If the overlap were in better 107 agreement with the observations, the cloud amount overestimation would be substantially 108 larger, such as that found in Fig. 3. 109 Thank you for this reference, this is exactly what we see in our model as well. We now 110 reference this paper. 111 112 Fig 4 & 5: the 2M lines are difficult to spot because all other lines are on top of it. It 113 might be worth it to replot this with 2M on top. 114 Done. 115 116 When I suggested to add the +/-1STD I meant annual STD, so that the frequency of 117 annual oscillation could be taken into account when compared to the model. I believe the 118 authors show here a monthly interannual STD rather than a yearly interannual STD. Id 119 recommend that the authors change the monthly interannual STD to yearly interannual 120 STD in these figures. I did it myself for the LW/SW CRE and the results is far smaller 121 than what shown here. 122 For LW CRE, SW CRE, cloud cover and precipitation the yearly interannual STD is too 123 small to be distinguished from the mean line (including line-width). It has therefore been 124 removed entirely. For the Li et al. (2012) dataset in Fig 5 the uncertainty provided by 125 the creators of the dataset is the most accurate description of uncertainty. We thus leave 126 the Figure as is. 127 128 P13 L13: it is not based 'on a threshold value for the backscattered polarization ratio to 129 separate...' but on the cross-polarized attenuated backscatter and the total attenuated 130 backscatter and a discrimination line. Note that this process is reproduced in the simulator 131

132 P13 L14-L15: The lidar spatial resolution is actually around 70m while the distance

- 133 between lidar shots is  $\sim$ 333m.
- 134 This aspects have been taken into account while rewriting Section 3.6.
- 135
- 136 P13 L33: They are more frequent in the 2M simulation than the REF and it should be
- 137 stated. While not comparable, it is also much more frequent in the observations.
- 138 This aspect has been taken into account while rewriting Section 3.6.
- 139
- 140 P14 L1: I don't understand this statement.
- 141 This does no longer exist in it's previous form.

142

P19 L17-18: This is wrong, it is due to comparing two different quantities, which do not take into account the instrument peculiarities. One should only conclude based on comparison between obs and model+sim outputs.

146 This sentence no longer exists.

147

- 148 P19 L18-19: As explained before, this is not whats shown in the paper.
- 149 This sentence no longer exists.

150

P19 L19-20: I completely disagree with this statement, which is not supported by any evidence. Even if it was shown that the PR-T definition is substantially different in ECHAM6 for thick and thin clouds (which is not performed so far), it would only suggest that the phase simulated with the lidar simulator cannot be used to assess the PR-T relationship of all clouds in this particular model.

156 This sentence no longer exists.

[revised manuscript text omitted]

---

## Author Response (AR3)

**Reply to anonymous Referee #1**

**Remo Dietlicher**

**June 18, 2019**

Thank you for going through the manuscript once more and catching a few last ambiguities.

**1  Specific Replies**

Line 10 p1: Underestimates the cloud phase partitioning? I guess the authors mean "underestimates the ratio of ice to liquid and ice cloud? (referred to as phase ratio in your manuscript). Please re-word so that the reader can understand without having to read the whole manuscript.

This is a good idea, we made it more clear.

Line 28 p12: There is no mixed-phase cloud category in the observations (only a likely mixed-phase cloud category) so it is not exactly true to say this. Maybe the authors should add something like "as well as the satellite product assuming that the undefined-phase clouds are considered as mixed-phase clouds (Cesana et al., 2016)".

This has been implemented.

Line 8 p13: "the one obtained by the simulator" → "than the observations".
Also maybe the authors should say explicitly that it means less ice clouds than in the obs.

You are right, this sentence did not make sense..

Line 17 p13: "COSP simulator" → "COSP simulator compared to the observations".

This has been implemented.

Last paragraph p13: It could also be that the model does not produce the ice formation below supercooled liquid clouds in the polar regions, which represent most of undef-phase clouds (likely mixed-phase clouds) in the observations.

A more detailed study on specific regions and causes for the lack of heterogeneous ice formation in these cold liquid clouds is needed. From Fig. 12 in the manuscript one can see, that Northern Hemisphere mid-latitudes and the Southern Ocean are a good place to start.

Or maybe it is classified as snow and snow is not considered as ice clouds in the 2M model compared to the default version if I understand correctly?

You understood correctly. However, snow primarily forms within *deep* clouds which usually cover the entire mixed-phase temperature regime, i.e. there is no effect from neglecting snowy levels. Levels that contain out-of-cloud snow that formed within mixed-phase clouds are rare.

[revised manuscript text omitted]